# Highly conserved shifts in ubiquitin-proteasome system (UPS) activity drive mitochondrial remodeling during quiescence

Sibiao Yue[1], Lei Wang[1], George N. DeMartino[1], FangZhou Zhao[1], Yi Liu [1] & Matthew H. Sieber [1] ✉

Defects in cellular proteostasis and mitochondrial function drive many aspects of infertility, cancer, and other age-related diseases. All of these conditions rely on quiescent cells, such as oocytes and adult stem cells, that reduce their activity and remain dormant as part of their roles in tissue homeostasis, reproduction, and even cancer recurrence. Using a multi-organism approach, we show that dynamic shifts in the ubiquitin proteasome system drive mitochondrial remodeling during cellular quiescence. In contrast to the commonly held view that the ubiquitin-proteasome system (UPS) is primarily regulated by substrate ubiquitination, we find that increasing proteasome number and their recruitment to mitochondria support mitochondrial respiratory quiescence (MRQ). GSK3 triggers proteasome recruitment to the mitochondria by phosphorylating outer membrane proteins, such as VDAC, and suppressing mitochondrial fatty acid oxidation. This work defines a process that couples dynamic regulation of UPS activity to coordinated shifts in mitochondrial metabolism in fungi, *Drosophila*, and mammals during quiescence.

Defects in mitochondrial metabolism and proteostasis underlie the onset and progression of numerous age-associated diseases such as reproductive dysfunction, neural degeneration, cardiovascular disease, and cancer[1–4,5–7]. A significant challenge in treating these conditions is understanding the mechanisms that coordinate dynamic changes in metabolism with global shifts in protein turnover. To address these questions, we utilize developmental systems, such as the transition between growth and quiescence, to understand how alterations in cellular metabolism are mechanistically linked to shifts in cellular proteostasis. Quiescent cells, such as adult stem cells and oocytes, display low levels of transcription and translation. However, once activated, these cells drive many aspects of development, wound repair, and gametogenesis[8–12].

An essential aspect of quiescence in systems ranging from fungal spores to mammalian adult stem cells is the suppression of metabolic activity in a process called mitochondrial respiratory quiescence (MRQ)[15,13,14]. MRQ supports this dormant state and protects quiescent cells from oxidative damage and other stresses until they are activated. In many of these quiescent cell populations, insulin signaling inhibits quiescence during growth. However, as cells enter quiescence, insulin/Akt signaling is suppressed and the critical regulator of MRQ, GSK3, becomes activated[13,15]. GSK3 induces a shift in the mitochondrial proteome that drives changes in mitochondrial morphology and suppresses mitochondrial oxidative metabolism (Fig. 1A). However, the mechanisms driving mitochondrial protein turnover in response to GSK3 are unknown.

Here we show that during cellular quiescence we have found that the proteasome is recruited to the mitochondria during quiescence to support the conserved suppression of mitochondrial respiration seen in many developmental systems. We have found that GSK3 regulates this

[1]Department of Physiology, UT Southwestern Medical Center, 5323 Harry Hines Blvd, Dallas 75390 TX, USA. ✉e-mail: matthew.sieber@utsouthwestern.edu

**Fig. 1 | The proteasome is recruited to the mitochondria during cellular quiescence. A** Model of Insulin-AKT-GSK3 regulation of mitochondrial respiration quiescence (MRQ). Created with Biorender. **B** Proteasome activity from st1–8 oocytes(active), st14 oocytes(quiescent), and embryos (collected 0–2h and 16–20 h posted egg laying) (*n* = 4 completely independent experiments on independent samples). **C** In-gel hydrolysis assay to examine total proteasome activity in st1–8 and st14 eggs. This experiment was replicated 3 times with biological replicates. **D** Western blotting examines K48 ubiquitin levels in total cell lysis in st1–8 and st14 eggs. **E** Mitochondria-associated proteasome activity in st1–8 and st14 oocytes (*n* = 6 independent biological replicates). **F** In-gel hydrolysis assay examine mitochondria-associated proteasome activity in st1–8 and st14 eggs. This experiment was replicated 3 times with biological replicates. **G** Western blotting to examine K48 ubiquitin levels in mitochondrial fractions in st1–8 and st14 eggs. This experiment was replicated 3 times with biological replicates. **H** Native Western blotting gels to examine 20 S core and 19 S regulatory components of the assembled proteasome in total cell lysis from st1–8 and st14 eggs. This experiment was replicated 3 times with biological replicates. **I** TMRE mitochondrial membrane potential staining of wild-type ovarioles treated with DMSO or proteasome inhibitor MG132 (50 μM, 2 h). All error bars represent 1 X standard deviation. **J** Summary of the effects of proteasome inhibitor MG132 (50 μM, 2 h) on TMRE mitochondrial membrane potential staining in the indicated stages. These data are normalized to the TMRE staining in the adjacent follicle cell (*n* = >20 biological replicates). In the box-and-whisker plots, the box represents the upper and lower quartiles and the line within the box is the mean. The whiskers represent the maximum and minimum values. *P* values were calculated by one-way analysis of variance (ANOVA). Error bars represent one standard deviation. **p < 0.001, *p < 0.05. Error bars represent 1X standard deviation.

recruitment of the proteasome to the mitochondrial by phosphorylating proteins in the mitochondrial outer membrane and suppressing mitochondrial fatty acid oxidation. In particular, we have found that phosphorylation of the mitochondria outer membrane ion channel VDAC plays a crucial role in proteasome recruitment and its physical association with the mitochondria. In addition, we have found proteasome recruitment to the mitochondria is highly conserved in systems from fungi, *Drosophila*, and mammalian cell models. Importantly,

proteasome recruitment to the mitochondria occurs in both developmentally-programed and stress-induced forms of quiescence.

## Results

### Proteasomes are recruited to the mitochondria during quiescence

To address this fundamental question of mitochondrial remodeling, we used *Drosophila* oocytes to examine the mechanisms that control

mitochondrial protein stability as cells enter quiescence. During oogenesis developing egg chambers are highly active and growing until stage 10 after which transcription and translation are suppressed and oocytes enters a state of complete quiescence in stage 14[8,9]. We discovered that, compared to actively growing oocytes (stage 1–8), the ubiquitin-proteasome system (UPS) activity increases 3-fold as oocytes enter cellular quiescence stage 14 (Fig. 1B). This UPS activity could be blocked using the proteasome inhibitor MG132 (Supplementary Fig. 1A). To confirm these findings we examined all three enzymatic activities of the proteasome (Trypsin-like, Chymotrypsin-like, and Caspase-like) using the proteasome activity probe Me4BodipyFL-Ahx3Leu3VS (UbiQ-018) (Supplementary Fig. 1G, H) and observed elevations in all three activities similar to what we observed with the AMC peptide-substrate proteasome activity assay.

Using our previously published RNA-Seq datasets that examine the changes in gene expression that occur as oocytes enter cellular quiescence[16], we observed a significant 1.4–2.5 fold increase in the mRNA expression of 26 genes involved with the proteasome and the UPS systems. These genes includes both 20 S core subunit factors and genes involved with the 19 S regulatory cap (Supplementary Data 1), suggesting that this increase in proteasome activity is driven by 26 S proteasome biosynthesis. Consistent with this, we found that the onset of quiescence in *Drosophila* oocytes is accompanied by elevated levels of the assembled 26 S proteasome (Fig. 1C), increased content of 26 S proteasome proteins (20 S, Rpt2, Rpt5, and Rpn12; Fig. 1H and Supplementary Fig. 1C), and modest changes in the levels of K48-linked ubiquitinated proteins (Fig. 1D). These data strongly suggest that quiescent oocytes have enhanced UPS function. To examine how this increased activity relates to mitochondria during quiescence, we purified these organelles from actively growing follicles and quiescent oocytes and measured proteasome activity. We found that our mitochondrial fractions were not contaminated with large amount of other membrane fractions such as the ER and plasma membrane (Supplementary Fig. 1B). At baseline in growing follicles, we found that mitochondria-associated proteasome activity contributes roughly 4% of total activity (Supplementary Fig. 1F). Unexpectedly, we discovered that mitochondrial fractions isolated from quiescent oocytes display a 12-fold increase in proteasome activity levels relative to mitochondria from growing follicles (Fig. 1E). We also observed increased levels of assembled 26 S proteasome and a dramatic elevation in K48 ubiquitinated proteins in mitochondria isolated from oocytes that have just entered quiescence, consistent with large amounts of active 26 S proteasome being recruited to the mitochondria during quiescence (Fig. 1F, G). Moreover, we observed that K48 ubiquitination returns to normal in the hours after the onset of quiescence when oocytes are stored suggesting that these ubiquitinated proteins have been turned over (Supplementary Fig.1 D). Interestingly, we observed stable levels of mitochondria-associated proteasome activity over this same period, suggesting that the presence of ubiquitinated protein substrates per se do not mediate this association between the mitochondria and the proteasome during quiescence (Supplementary Fig. 1 D, E). These data are consistent with the known the roles of the UPS in mitochondrial protein quality control and maintaining proper respiration and the known associations with UPS components with the mitochondria[17,18]. In light of these studies, our data suggests that proteasome recruitment may be fundamental connection between mitochondrial metabolism and the pathways that facilitate cytosolic protein turnover in many contexts. To gain insight into how proteasome function impacts mitochondrial activity during oogenesis, we acutely treated dissected ovaries with the proteasome inhibitor MG132. Using this system, we observed that suppressing proteasome function during oogenesis leads to significant increases in mitochondrial membrane potential at several developmental stages (Fig. 1I, J). These data are consistent with proteasome recruitment being linked to mitochondrial remodeling during quiescence.

## GSK3 regulates proteasome recruitment to the mitochondria during quiescence

Given that GSK3 is a crucial regulator of mitochondrial remodeling during MRQ[15], we hypothesized that mitochondria-associated proteasomes might be a key mechanism by which GSK3 drives mitochondrial remodeling. We tested this model by measuring proteasome activity in mitochondrial fractions and GSK3-RNAi quiescent oocytes. display Compared to mitochondria from untreated controls, mitochondria from GSK3-RNAi oocytes display reductions in proteasome activity, 26 S proteasome content, and K48 polyubiquitylated proteins (Fig. 2A). In line with previous data[15], we observed that GSK3-RNAi oocytes display elevated mitochondrial respiration and persistently high levels of mitochondrial membrane potential in stage 10 follicles, consistent with a block in MRQ (Fig. 2B & Supplementary fig 2A). We also found that GSK3-RNAi oocytes display reduced levels of stored triglycerides and acyl-carnitines consistent with high mitochondrial fatty acid oxidation (FAO) (Fig. 2C & Supplementary Fig. 2B). Previous proteomic studies showed that over 60 mitochondrial proteins are reduced in abundance in mitochondria from quiescent oocytes[15]. When we examined FAO proteins in the mitochondria from quiescent oocytes, we discovered several key proteins involved in mitochondrial FAO, including ACAA, ETFα, ETF-QO, and MTPα, are reduced in mitochondria from quiescent cells (Fig. 2D). Interestingly, silencing GSK3 expression prevents the loss of ACAA and ETFα, consistent with elevated levels of mitochondrial FAO in quiescent GSK3-RNAi oocytes (Fig. 2D). Taken together these data suggest that GSK3 inhibition of FAO is linked to proteasome recruitment during MRQ.

Consistent with this model, inhibiting the mitochondrial FAO genes ETFA and MTPα caused a reduction in both mitochondrial membrane potential and respiration (Fig. 2E–G & Supplementary Fig. 2C). Intriguingly, when we measured proteasome activity in mitochondria isolated from ETFA-RNAi and MTPα-RNAi follicles, we discovered that inhibition of mitochondrial FAO caused increased levels of mitochondria-associated proteasome activity (Fig. 2H, I) and K48 poly-ubiquitination. These data suggest that during quiescence GSK3 regulates the stability of mitochondria fatty acid oxidation proteins and, in turn, suppression of fatty acid oxidation promotes proteasome recruitment to the mitochondria. We excluded the possibility that compromised mitochondrial respiration induces proteasome recruitment to the mitochondria by inhibiting ETC activity, by treating cells with either complex 1 inhibitor (0.3 uM rotenone) or complex III inhibitor (0.3 Antimycin A) for 6 h, and assaying proteasome activity in mitochondrial fractions. In both experiments ETC inhibition had no effect on proteasome recruitment to the mitochondria (Supplementary Fig. 2F). It is important to note that due to differences in metabolic state that are present between various tissues and cell lineages the effect of FAO inhibition on proteasome recruitment to the mitochondria in other quiescent cell populations such as HSCs and muscle satellite cells is unclear.

## GSK3 triggers the turnover of outer membrane proteins

To define the mechanism for GSK3 of regulation of proteasome recruitment to the mitochondria, we examined GSK3 localization using cellular fractionation (Fig. 3A). We discovered that a significant portion of GSK3 is present in mitochondrial fractions from quiescent oocytes, consistent with the idea that GSK3 phosphorylates targets in the outer mitochondrial membrane to induce MRQ (Supplementary Fig. 2D). To test this model using an unbiased systematic approach, we combined proximity labeling with cellular fractionation to identify the mitochondrial-specific targets of GSK3. We constructed GSK3-APEX2 transgenes and expressed them in the germ cells of the *Drosophila* ovary. We then confirmed that the transgenes express at significant levels, that we could detect intense biotin labeling of mitochondria,

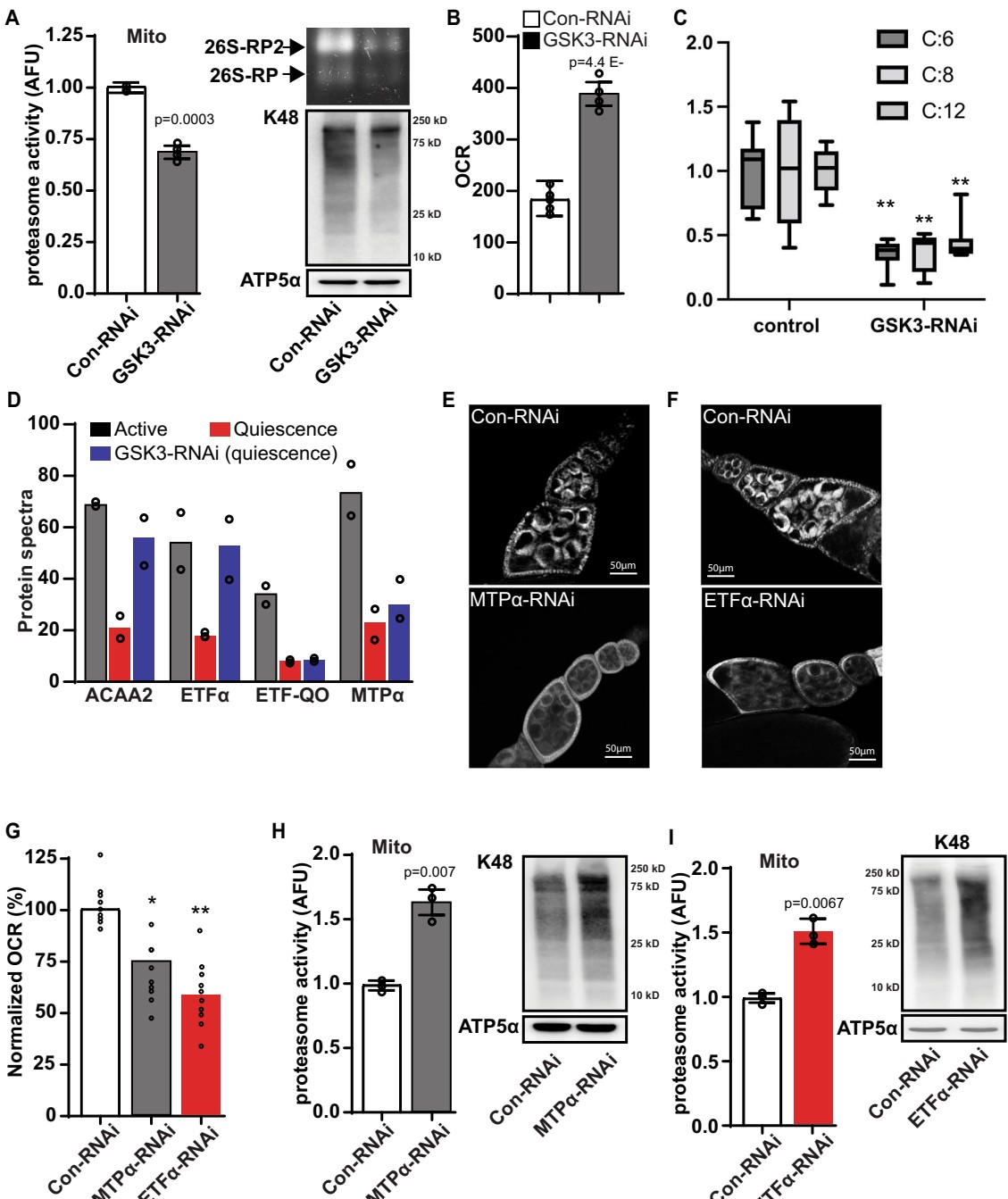

**Fig. 2 | GSK3 mediate suppression of fatty acid oxidation triggers proteasome recruitment. A** Mitochondria-associated proteasome activity, proteasome zymography, and K48 ubiquitin levels in mitochondrial fractions in st14 eggs from control-RNAi and GSK3-RNAi ovarioles. (4 replicate experiments on independent biological replicates). **B** The oxygen consumption rate (OCR) measurements in st14 eggs from control-RNAi and GSK3-RNAi ovarioles ($n = 6$ biological replicates). **C** Acyl-carnitine levels from LC/MS analysis of st14 eggs from control-RNAi and GSK3-RNAi ovarioles($n = 8$ biological replicates). In the box-and-whisker plots, the box represents the upper and lower quartiles and the line within the box is the mean. The whiskers represent the maximum and minimum values. **D** Levels of enzymes involved in mitochondrial fatty acid oxidation pathway (FAO) from proteomic analysis of eggs from st1–8 (active growth), st14 (control-RNAi), and st14 (GSK3-RNAi) ovarioles ($n = 2$ biological replicates). **E, F** TMRE mitochondrial membrane potential staining of control-RNAi, MTPα-RNAi(E), and ETFα-RNAi (F) ovarioles. Quantification in Figure S2. **G** OCR measurements in st10 eggs from a control-RNAi, MTPα-RNAi, and ETFα-RNAi ovariole($n = 9$ biological replicates). Mitochondria-associated proteasome activity ($n = 3$ independent experiments on independent biological samples) and K48 ubiquitin levels in mitochondrial fractions from st14 eggs from control-RNAi, MTPα-RNAi (**H**), and ETFα-RNAi (**I**) ovarioles. Significance values are calculated by a two-tailed student's $t$-test assuming unequal variance. Error bars represent one standard deviation. $^{**}p < .001$, $^{*}p < .05$. Error bars represent 1 X standard deviation.

and validated our ability to efficiently pulldown GSK3-associated mitochondrial proteins (Supplementary Fig. 3 A–C). We utilized this system to examine the mitochondrial proteins that associate with GSK3-APEX2 relative to CD8-APEX2 controls[19]. Using a logFC cut-off of

1.0 (GSK3-APEX2/CD8-APEX2), of the we identified 57 well known mitochondrial proteins associated with GSK3(Supplementary Data 2). Gene ontology analysis of these proteins shows they are highly enriched for processes such as respiration, mitochondrial transport, and

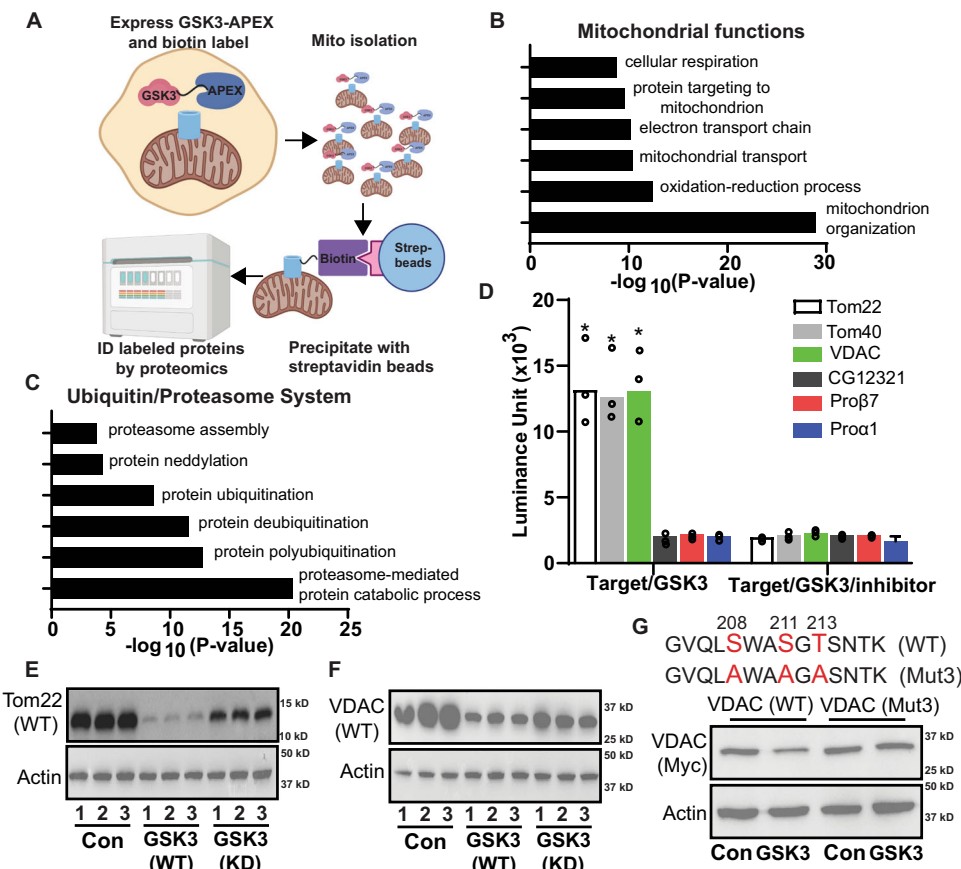

**Fig. 3 | GSK3 phosphorlaytes mitochondrial outermembrane proteins and controls their turnover. A** Experimental strategy to identify mitochondrial GSK3 interacting partners during quiescence. Created with Biorender. GO term analysis of proteins involved in mitochondrial functions (**B**) and Ubiquitin/proteasome system (**C**) identified by GSK3-Apex2 proximity labeling. **D** In vitro kinase assay tests GSK3 candidate targets identified in (**B**) (Tom complex components and VDAC) and (**C**) (UPS components) ($n$ = 3 experimental replicates). Validation of GSK3 Kinase targets TOM22 (**E**) and VDAC (**F**) in 293 T cells ($n$ = 4 experimental replicates). **G** A diagram of the of phosphorylation sites (Red letters) essential for GSK3 regulation of VDAC protein levels and a western examining how the mutation of these sites impact GSK3 mediated turnover of VDAC. Lanes 1 and 2 show the impact of wt GSK3 expression on VDAC stability. Lanes 3 and 4 shows phosphorylation site mutations in block VDAC GSK3 induced VDAC turnover($n$ = 3 experimental replicates). Significance values are calculated by a two -tailed student's $t$-test assuming unequal variance. Error bars represent one standard deviation.

mitochondrial organization (Fig. 3B), consistent with data showing GSK3 suppresses mitochondrial oxidative metabolism and drives mitochondrial remodeling during MRQ.

We also identified 73 GSK3-associated proteins involved with the ubiquitin-proteasome system, including subunits of the 20 S core particle, 19 S regulatory particles, proteasome assembly factors, and proteins involved with ubiquitination and de-ubiquitination (Fig. 3C) (Supplementary Data 3). Of the 609 proteins identified in this approach 130 have well established functions as mitochondrial specific proteins or components of the UPS many of the other proteins are found in multiple cellular compartments (including mitochondria) or have unknown cellular localizations. These data displayed a high degree of reproducibility and a strong correlation between samples (Supplementary Fig. 3D–G). These data are consistent with the ability of GSK3 to regulate MRQ by promoting the recruitment of the proteasome to the mitochondrial surface.

We hypothesized that GSK3 might phosphorylate target proteins in the outer mitochondrial membrane to induce proteasome recruitment and mitochondrial protein turnover. To test this hypothesis, we selected VDAC and the Tom complex (Tom22 and Tom40) from our dataset as ideal candidate targets of GSK3. Using in vitro kinase assays, we discovered that purified recombinant GSK3 could directly phosphorylate Tom22, Tom40, and VDAC at high levels (Fig. 3D). Moreover, their phosphorylation by GSK3 can be blocked by adding a GSK3 inhibitor to the reaction (Fig. 3D).

In contrast, the proteasome proteins Prosβ7, Prosα1, and CG12321/PSMG2 that were also present in our GSK3-APEX2 data set cannot be phosphorylated by GSK3(Fig. 3D). These data suggest that their enrichment in our data set likely reflects the proteasome recruitment to the mitochondria in response to GSK3-mediated phosphorylation of mitochondrial outer membrane proteins. Using a 293 T cell co-transfection system, we examined how phosphorylation by GSK3 impacts the stability of recombinant Tom22 and VDAC. We found that expressing GSK3 in 293 T cells causes a reduction in both Tom22 and VDAC consistent with their turnover (Fig. 3E, F & Supplementary Fig. 4A, B). In contrast, the expression of a kinase-dead version of GSK3 (GSK3-KD) suppressed the turnover of these proteins. Although two of three candidate phosphorylation sites in the VDAC coding sequence are not required for the GSK3-mediated turnover of VDAC by GSK3 (Supplementary Fig. 4C), the SWASGT site located at amino acid 208–213 is required for this response (Fig. 3G & Supplementary Fig. 4C, D). These data suggest that GSK3 promotes proteasome recruitment to the mitochondria and, once recruited, the proteasome degrades mitochondrial proteins such as VDAC and Tom22.

When we inhibited Tom22 in *Drosophila* germ cells using RNAi, we found that Tom22-RNAi ovaries display only a pair of vasa-staining germ cells at each ovariole base (Supplementary Fig. 5A). Over two weeks, those cells were lost (Supplementary Fig. 5A). This observation is consistent with the half-life of the *Drosophila* germline stem cell.

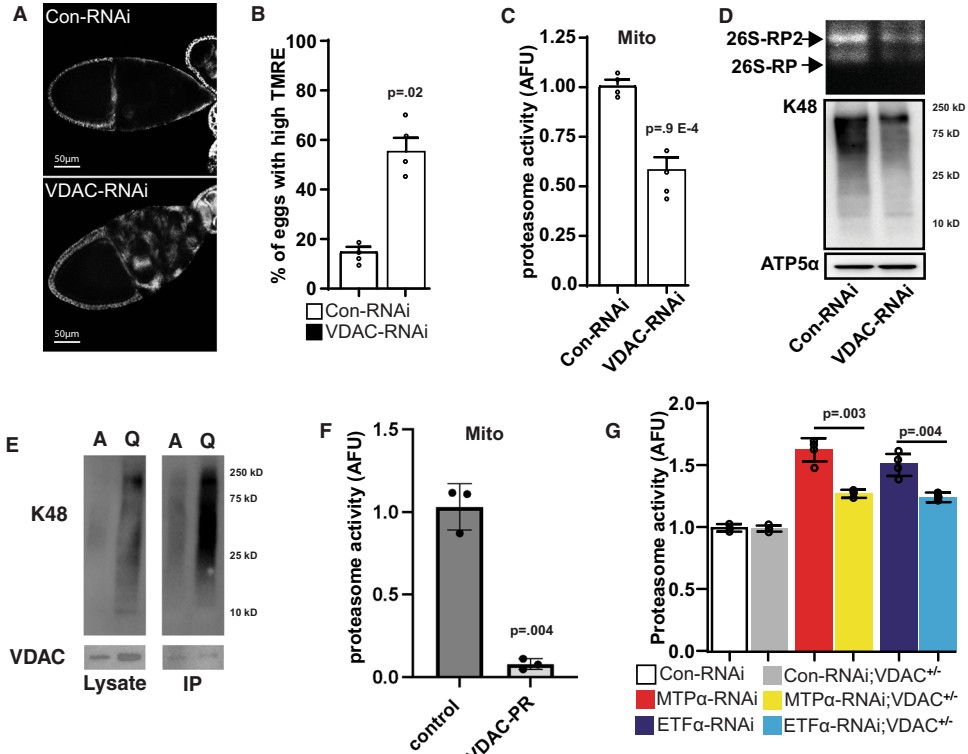

**Fig. 4 | GSK3-mediated phosphorylation of VDAC drives proteasome recruitment. A** TMRE mitochondrial membrane potential staining of st10 eggs from a control-RNAi and VDAC-RNAi ovariole. Quantification in Fig. S6A. **B** Quantification of the number of stage 10 egg chambers that display high levels of the TMRE staining ($N = 20$ ovaries). **C** Mitochondria-associated proteasome activity in mitochondrial fractions from st14 eggs from control-RNAi and VDAC-RNAi ovariole ($n = 4$ experiments on independent biological replicates). **D** K48 ubiquitin levels and zymography in mitochondrial fractions from st14 eggs from control-RNAi and VDAC-RNAi ovariole ($n = 4$ experiments on independent biological replicates).

**E** K48 ubiquitination assay for VDAC in active and quiescent oocytes ($n = 3$ independent experiments). **F** Mitochondrial associated proteasome activity from control oocytes and oocytes that overexpress a Phospho-resistant form of VDAC (VDAC-PR) ($n = 3$ independent experiments). **G** Epistasis analysis of VDAC1 and FAO pathway in regulating mitochondria-associated proteasome activity ($n = 5$ biological replicates). *P* values were calculated by one-way analysis of variance (ANOVA). Error bars represent one standard deviation. **p** < 0.001, *p* < 0.05. Error bars represent 1 X standard deviation.

---

However, due to the severe developmental phenotypes we observed with Tom22-RNAi, we focused our remaining studies on VDAC.

## GSK3 phosphorylation of VDAC promotes proteasome recruitment

Inhibition of VDAC in the germline resulted in stage 10 egg chambers maintaining their membrane potential similar to oocytes with inactive GSK3 (Fig. 4A & Supplementary Fig. 6A). VDAC inhibition causes a minor reduction in the levels of oxygen consumption and elevated levels of stored glycogen (Supplementary Fig. 6B, C). These data are consistent with the reduced complex I activity previously observed in VDAC mutants[20] and suggest that mitochondrial membrane potential can be uncoupled from respiration during MRQ. Furthermore, our metabolomic analysis of VDAC-RNAi oocytes revealed a slightly altered metabolic signature, including some modest effects on FAO-related metabolites (3-hydroxysebacic acid, tetradecanoylcarnitine[21]) in VDAC-RNAi oocytes (Supplementary Fig. 6D–F). Importantly, mitochondria purified from VDAC-RNAi oocytes display a ~40–50% reduction in mitochondria-associated proteasome activity (Fig. 4C), content, and K48-poly-ubiquitination, feature that phenocopy the effect of GSK3 inhibition (Fig. 4D). Using ubiquitination assays we found that VDAC displays high levels of ubiquitination in quiescent oocytes (Fig. 4E). We hypothesized VDAC may function directly in the recruitment of the proteasome and to test this we purified proteasomes from quiescent cells using a RPT6-Flag expressing cell line and examined whether mammalian VDAC1 was associated with the proteasome. Interestingly, compared to the control Co-IP, VDAC1 show a significantly enriched association with the proteasome

(supplementary Fig. 5C), compared to the control IP, suggesting VDAC functions as an important cue that triggers proteasome recruitment to the mitochondria. Consistent with these results GSK3 phosphorylation of VDAC displays a much milder effect on VDAC turnover than what we observe with TOM22. Taken together these data suggest during quiescence GSK3 triggers proteasome recruitment and VDAC functions to recruit the proteasome to the surface via a potential direct interaction. Once recruited to the mitochondrial surface a subset of VDAC, possibly unbound by the proteasome, is turned over by the UPS. These data suggest that VDAC phosphorylation plays a crucial role in the recruitment of the proteasome to the mitochondria. We tested this model by overexpressing a phosphor-resistant version of VDAC (VDAC-PR) conditionally during oogenesis. Interestingly, VDAC-PR expression was sufficient to block the recruitment of the proteasome to the mitochondria observed in quiescent oocytes (Fig. 5F) suggesting VDAC phosphorylation is a key signal that triggers proteasome recruitment during MRQ. In addition, these data suggest that VDAC-PR overexpression has a dominant-negative effect on proteasome recruitment. Given our data shows that GSK3 regulates VDAC and fatty acid oxidation during MRQ, we hypothesized that these factors might function together as part of a common regulatory mechanism downstream from GSK3 to recruit the proteasome to the mitochondria during quiescence. To test this hypothesis, we introduced a heterozygous null mutant of VDAC into the MTPα-RNAi and ETFA-RNAi lines and measured mitochondria-associated proteasome activity in germ cells. Interestingly, this was sufficient to suppress the increased levels of mitochondria-associated proteasome activity and K48 ubiquitination observed in both MTPα-RNAi and ETFA-RNAi

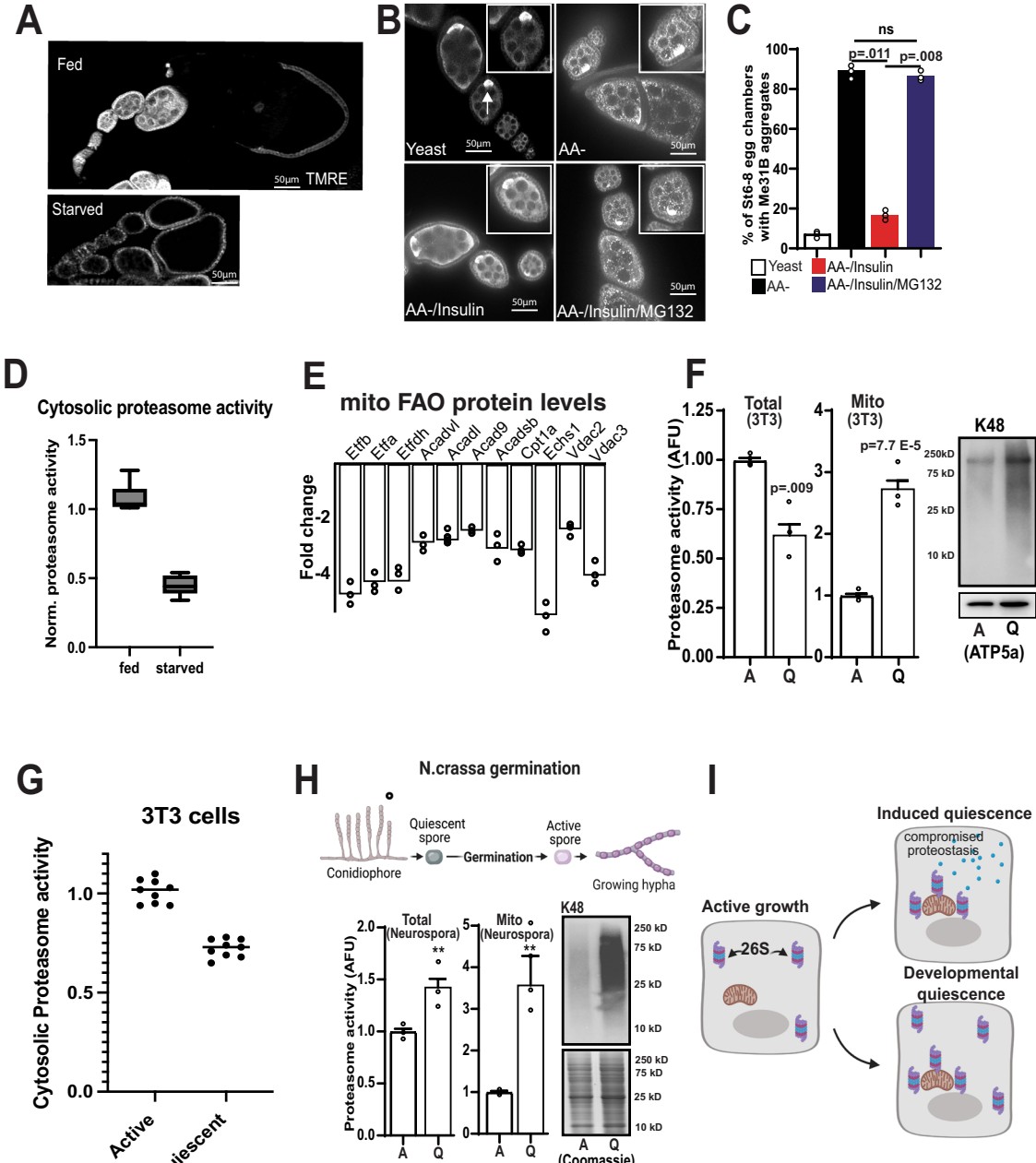

**Fig. 5 | Proteasome recruitment to the mitochondria is highly conserved in developmental and starvation-induced forms of quiescence. A** TMRE staining of ovarioles from starved vs fed animals. **B** Immunostaining wild-type ovarioles to detect Me31B with indicated feeding and treatment strategies. **C** Summary of percentage of stage 6–8 egg chambers with Me31B aggregates in **B** ($n = 25$ egg chambers). *P* values were calculated by one-way analysis of variance (ANOVA). Error bars represent one standard deviation. **D** Cytosolic proteasome activity from stage 1–8 egg chambers from fed and starved oocytes ($n = 3$ completely independent experiments on independent samples 6 total samples). **E** Fold changes of selected FAO pathway enzymes and VDACs in quiescent NIH 3T3 cells compared with active ones ($n = 3$ independent replicates). **F** Total and mitochondria-associated proteasome activity and K48 ubiquitin levels in actively growing and quiescent NIH 3T3 cells($n = 3$ independent experiments on multiple samples 7 total samples). **G** Cytosolic proteasome activity from active and quiescence NIH 3T3 cells ($n = 3$ completely independent experiments on independent samples (9 samples total)). **H** Total and Mitochondria-associated proteasome activity and K48 ubiquitin levels in quiescent and germinated *N.crassa* spores ($n = 3$ completely independent experiments on independent samples). **I** Model of how the proteasome participates in either induced or developmental quiescence. Created with Biorender. In the box-and-whisker plots, the box represents the upper and lower quartiles and the line within the box is the mean. The whiskers represent the maximum and minimum values. Significance values are calculated by a two-tailed student's *t*-test assuming unequal variance. Error bars represent one standard deviation. **p < 0.001, *p < 0.05.

follicles (Fig. 4G and Supplementary Fig. 6G). These data indicate that VDAC aides in the recruitment of the proteasome to the mitochondria and promotes the turnover of mitochondrial outer membrane proteins. Moreover, our data show that VDAC is a crucial factor acting downstream of GSK3 and fatty acid oxidation in recruiting the proteasome to the mitochondria during MRQ (Supplementary Fig. 5B).

**Proteasome recruitment is conserved in developmental and induced forms of quiescence**

In addition to developmental regulation, quiescence can also be induced by nutrient deprivation[22]. Feeding females an amino acid-deficient diet causes a precocious onset of MRQ during oogenesis (Fig. 5A)[15]. At the same time, it has been shown that several meta-stable

mRNA binding proteins, such as Me31B, undergo a phase shift and form aggregate puncta under these conditions[23]. We thus hypothesized that recruiting the proteasome to the mitochondria during this induced form of MRQ may contribute to the Me31B aggregates seen in amino acid-deficient conditions. In line with this hypothesis, we found that while adding exogenous insulin was sufficient to clear the Me31B aggregates, adding back insulin in the presence of proteasome inhibitor blocked clearance of Me31B aggregates (Fig. 5B, C). These data suggest that cytosolic proteasome levels are a major factor in Me31B aggregation and suggest that cytosolic proteasome activity is compromised in induced forms of quiescence. Given that this starvation-induced form of MRQ occurs before the 3-fold increase in proteasome activity observed during developmental MRQ, these data suggest that proteasome recruitment to the mitochondria during induced quiescence compromises cytosolic proteostasis and contributes to protein aggregation (Fig. 5D). Consistent with this model we measured cytosolic proteasome activity in developing egg chambers from fed and amino-acid starved females and found a roughly 50–60% reduction in cytosolic proteasome activity consistent with the increased aggregation of mei31B further supporting the idea that induced forms of quiescence display compromised proteostasis. Moreover, this work is supported by recent studies that have suggested a direct link between mitochondrial metabolism and UPS regulation[24,25].

To test the model that induced forms of quiescence display increased mitochondria-associated proteasome activity at the expense of cytosolic activity, we examined a mouse fibroblast model of induced quiescence[26–28]. We confirmed that inducing quiescence in mouse 3T3 cells causes the same reduction in mitochondrial respiration and glycolysis that we observed in *Drosophila* oocytes. We also observed a similar increase in-stored glycogen in quiescent fibroblasts (Supplementary Fig. 7A–D). An examination of the mitochondrial proteome of quiescent fibroblasts revealed that 93 mitochondrial proteins display reduced levels during quiescence, similar to findings in *Drosophila*[13,15] (Supplementary Data 4). These proteins are enriched in gene ontology groups that include transit peptides, the mitochondrial inner membrane, mitochondrial outer membrane, acetylation, TCA cycle, and carbon metabolism (Supplementary Fig. 7E). Within this data set, we also observed a decrease in the ETF complex levels, ACAD enzymes, and two VDAC family members in mitochondria from quiescent fibroblasts, similar to our results from *Drosophila* (Fig. 5E). These data that MRQ is conserved from flies to mammals and that this system represents a viable tool to examine proteasome recruitment to the mitochondria during quiescence.

Consistent with our induced quiescence model, mitochondria-associated proteasome activity and K48 ubiquitination increase dramatically during quiescence in fibroblasts (Fig. 5F). However, in this induced form of quiescence, total proteasome activity is significantly lower and proteasome recruitment to the mitochondria further compromises cytosolic protein turnover rate (Fig. 5F). To confirm these findings we examined all three enzymatic activities of the proteasome (Trypsin, Chymotrypsin, and Caspase) using the proteasome activity probe Me4BodipyFL-Ahx3Leu3VS (UbiQ-018) (Supplementary Fig. 1F, G) and observed reduced levels of all three activities similar to what we observed with the AMC-substrate proteasome assay. Consistent with this idea we observed a roughly 30% decrease in cytosolic proteasome activity in quiescent 3T3 cells very similar to what we observed in oocytes from A.A. starved female flies (Fig. 5G). To examine the differences between developmental quiescence and induced quiescence, we used a well-established model of developmental quiescence, the germination of the asexual spores of the filamentous fungus *Neurospora crassa*. Just as in animals, the *N.crassa* asexual spores display suppressed mitochondrial activity during quiescence[29–35]. As expected, we found that both total proteasome activity and mitochondria-associated proteasome activity are significantly elevated during quiescence in this system (Fig. 5G, H). Thus, the increase in total

proteasome activity during developmental quiescence helps maintain cytosolic proteostasis during quiescence when the proteasome is recruited to the mitochondria during MRQ (Fig. 5I). Moreover, our data indicate that proteasome recruitment to the mitochondria is widely conserved from fungi to mammals and represents a fundamental aspect of quiescence which is associated with the suppression of mitochondrial function in quiescent cells.

## Discussion

In summary, our work has identified an evolutionarily conserved relationship between dynamic changes in cellular proteostasis and the remodeling of mitochondrial metabolism during cellular quiescence. As cells enter quiescence, proteasome activity and content increase, and the 26 S proteasome is recruited to the mitochondria. This enhanced mitochondria-associated proteasome activity supports the significant shift in the mitochondrial proteome we observed that underlies the onset of MRQ. We found that in *Drosophila* GSK3 functions as a critical regulator of proteasome recruitment to the mitochondrial outer membrane by directly phosphorylating outer-membrane proteins, particularly Tom22 and VDAC, and targeting them for degradation. GSK3 also promotes proteasome recruitment by inducing the turnover of components of the mitochondrial fatty acid oxidation machinery and inhibiting fatty acid catabolism. Interestingly, suppressing fatty acid oxidation recruits the proteasome in a VDAC-dependent manner suggesting that VDAC is a fundamental factor in proteasome recruitment and the induction of MRQ. We believe that VDAC may function as a direct anchor point for proteasome recruitment and that once recruited to the mitochondrial surface VDAC not bound by the proteasome is polyubiquitinated and degraded. This is supported by our data which shows GSK3 phosphorylation is required for proteasome recruitment to the mitochondria (Fig. 4C, F) and our data that VDAC display an enriched association with the proteasome in Co-IP experiments from quiescent cells. Once bound to the mitochondria we believe the proteasome may support mitochondrial remodeling by degrading proteins subject to retrograde transport, by blocking mitochondrial protein transport (via TOM complex turnover), or by disrupting the mitochondrial outer membrane leading to activation of a mitochondrial protease cascade. Overall, our data provide an intriguing mechanism for how disruptions in mitochondrial metabolism can directly regulate cellular proteostasis through proteasome recruitment to the mitochondria.

Our studies also highlight how different forms of quiescence can shift this relationship from being protective to being deleterious. We found that the proteasome is recruited to the mitochondria during developmental and starvation-induced forms of quiescence. However, while total cellular proteasome activity increases during developmental quiescence, starvation-induced forms of quiescence display lower levels of cytosolic activity. Thus, the proteasome's recruitment to mitochondria during induced quiescence contribute to the impairment of cytosolic protein turnover and highlights how changes in mitochondrial pathways, such as fatty acid oxidation, can compromise cytosolic proteostasis in quiescent cells by inducing proteasome recruitment to the mitochondria.

Interestingly, studies have shown that inducing spore formation in aged yeast can reverse the aging process and facilitate the clearance of age-associated protein aggregates[36]. Our data suggest that enhanced UPS activity and its regulation of mitochondrial function may contribute to aspects of this reversal of cellular aging. Moreover, recent work has suggested that mitochondria play a direct role in cytosolic protein turnover and protein aggregates clearance[37,38]. Our data suggest that proteasome recruitment to the mitochondria provides a mechanistic framework that may explain aspects of how mitochondria contribute to cytosolic protein turnover. While we have examined this mechanism in germ cells, in vitro cell culture systems, and fungal spores, this enhancement of proteasome activity and

 

recruitment to the mitochondria may drive the remodeling of metabolism and proteostasis in other quiescence cell populations such as adult stem cells (muscle satellite cells and hematopoietic stem cells, for example). As a result, corruption of this mitochondria/UPS relationship may drive the stem cell-associated defects that underlie the decline in wound repair and compromised immune response observed in diabetic patients and people suffering from metabolic syndrome[39–42].

Intriguingly, recent work has implicated VDACs in the etiology and progression of Alzheimer's Disease (AD). They are thought to provide a functional link between the aberrant protein aggregation and mitochondrial dysfunction that underlie neural degeneration in this disease[5–7,43,44]. The role of VDAC in recruiting the proteasome to the mitochondria provides a possible mechanism for how defects in mitochondrial function may enhance protein aggregation in AD and other degenerative conditions. Moreover, this role VDAC coordinating mitochondrial metabolism and cellular proteostasis provides a potential target for therapeutic approaches to prevent mitochondrial dysfunction and protein aggregation seen during aging and neural degeneration.

## Methods

### Experimental models and subject details

**Drosophila stocks**. *Drosophila melanogaster* stocks were kept on cornmeal molasses food at room temperature prior to the experimental dietary treatments outlined below. This food was prepared by adding 67 g/L cornmeal, 6.7 g/L agar, 25 g/L brewer's yeast, and 84 ml/L molasses to boiling water and boiling for 20 min. After cooling, 10 ml/L tegosept solution and 5 ml/L propionic acid were added before pouring food into plastic vials or bottles. All stocks were obtained from the *Drosophila* Bloomington stock center.

Adult female flies, 5–7 days old, were used for all experiments unless otherwise specified in the figure legends. Adult flies were matured on media supplemented with fresh yeast paste for at least 2 days.

### Normal and modified dietary (amino acid deficiency) feeding experiments

To ensure reproducibility, we used an amino acid-deficient grape agar media previously published to suppress insulin release from the IPCs. This media is commercially available from Genesee Scientific (cat# 47–102). The control diet is a standard *Drosophila* culture medium.

### Cell lines and plasmid transfection

NIH3t3 cells (ATCC CRL-1658) and Hek293T (ATCC CRL-3216) cells were purchased directly from ATCC (01/2020). 3T3 cells were validated by their ability to differentiate into myotubes (04/2022). HEK293T cells were authenticated by resSeq-Q-PCR assay (04/2022). 293T and NIH 3T3 cells were cultured in DMEM (+10% FBS) at a temperature of 37 °C with 5% CO2. For transfection, 293T cells were first seeded in 24-well plates one day before and reached ~70–90% confluent on transfection day. ~500 ng target plasmids and ~500 ng control or GSK3 plasmids were co-transfected using TurboFect™ Transfection Reagent (Thermo # R0532). The culture medium was replaced 24 h after transfection, and samples were directly lysed in 1 X SDS sample buffer 48 h after transfection.

### *N. crassa* culture conditions

For *Neurospora crassa*, quiescent spores were obtained from FGSC4200 strain grown on minimal slants (3% sucrose, 1.5% agar, 1x Vogel's salt solution[45]) for 8–9 days. The spores were resuspended and washed 3 times using 1 M sorbitol. To obtain germinating conidia as active spores, sufficient conidia from FGSC4200 strain grown on minimal slants in two 250 ml flasks for 8 days were resuspended and inoculated in 1 L liquid medium (2% sucrose, 1x Vogel's salt solution). The conidia were allowed to germinate on a shaker for 6 h at room temperature and then dried and collected.

### cDNAs

The cDNAs used in our molecular studies (Tom40 clone# 1609802, CG12321 clone# 1617048, Prosbeta7 clone# 1616658, Prosalpha1 clone# 1617072, Tom22 clone# 0180945, VDAC clone# 0137243 and GSK3 clone# LD44595) were obtain from the *Drosophila* genome resource center (DGRC).

### Primers

(See Supplementary Data 5)

### TMRE staining

Dissected ovarioles were incubated in 1xPBS (10 μg/ml bovine insulin) using 10 nM freshly prepared TMRE for 15 min with gentle agitation every 3 min. Ovarioles were then washed 3 times in 1xPBS (10 μg/ml bovine insulin) and mounted on slides in 1xPBS (10 μg/ml bovine insulin). The oocytes were imaged immediately using ApoTome.2 fluorescence microscope. For testing effects of blocking proteasome function on mitochondrial membrane potential, we modified the original method and excluded any effect of exogenous insulin during experiments. Dissected ovarioles were incubated in 1xPBS (no addition of insulin) with either DMSO or 50 μM MG132 for ~1.5–2 h. The ovarioles were then washed 3 times in 1xPBS before performing TMRE staining and imaging following the standard protocol (no addition of insulin). All TMRE data is quantified based on pairwise comparisons with controls grown on the same food and stained on the same day.

### Construct cloning and protein expression

For mammalian expression assay, the cDNA of *Drosophila* GSK3 and its candidate targets with either Myc or Flag sequence added before the stop exon were cloned downstream from the CMV promoter in the pcDNA3.1-His-A plasmid. To clone kinase-dead GSK3, *Drosophila* GSK3 protein sequences 83K84K were mutated to 83M84A. The Primer pairs with mutated nucleotides were used to introduce 83M84A in WT GSK3 sequence and used to synthesize two overlap PCR fragments containing mutated sites following the standard PCR protocol. After gel purification, the two PCR fragments were annealed and used to synthesize the full GSK3 PCR product containing 83M84A mutations following the standard PCR protocol.

For cloning the phosphorylation site mutations of VDAC, serine, or tyrosine of predicated sites were mutated to alanine. The same PCR protocol for cloning kinase dead GSK3 were used for VDAC experiments, and the primer pairs to introduce mutation sites were included in supplementary table. The DNA sequences of all plasmids were confirmed by sequencing, and the expression of these plasmids in 293 T cells was validated before experiments. For expression of recombinant VDAC in bacteria, the cDNA of Drosophila VDAC was cloned downstream from the T7 promoter of the pET-28a (+) plasmid. The expression of recombinant VDAC was induced at 37 C for 3 h by adding 0.1 mM IPTG to cultured BL21 cells. Inclusion bodies containing recombinant VDAC were first dissolved in 6 M Guanidine hydrochloride and then purified by NEBExpressR Ni Spin Columns (NEB, #1427 S). The refolding of purified VDAC protein was performed by multiple rounds of dialysis and then analyzed by SDS-PAGE.

GSK3-apex was made by subcloning GSK3 into pUC119. We then subcloned APEX into pUC119 to generate a C-terminal fusion with GSK3 and the FLAG epitope. The resulting fusion transgene was then subcloned into the Drosophila pPW expression vector via directional gateway cloning. (primer sequences are located in Supplementary Data 5).

## Mitochondrial isolation

The ovaries of females of the corresponding genotype were dissected. Ovaries were dissociated in Grace's media, and ~100 mg of specifically staged oocytes were isolated and then rinsed twice in mitochondrial isolation buffer (0.25 M sucrose, 1 mM EDTA, 100 µM ATP and 10 mM Tris-HCl (pH 7.1)). The oocytes were next homogenized in 200 µL of ice-cold mitochondrial isolation buffer. The ~170 µL supernatant was transferred to a new tube containing 830 µL mitochondrial isolation buffer and centrifuged at $1000 \times g$ for 10 min twice to remove nuclei and cell debris. The supernatant was transferred to a new microcentrifuge tube and centrifuged at $8000 \times g$ for 15 min yielding a crude mitochondrial pellet. The resulting pellet was then resuspended in at least 800 ul isolation buffer and centrifuged at $10,000 \times g$ for 8 min twice to yield mitochondrial pellet for subsequent experiments. To examine mitochondria-associated proteasome activity, we choose a mitochondrial isolation buffer (0.25 M sucrose, 20 mM Tris-HCl (pH 7.5), 20 mM NaCl, 5 mM MgCl2, 2 mM 2-Mercaptoethanol, 100 µM ATP) compatible for proteasome activity assay. To isolate mitochondria from Neurospora crassa, ~200 mg wet quiescent spores or active spores were first dissolved in 300 µl mitochondrial isolation buffer and mixed with 100 µl Zirconium beads (800 µm diameter). The samples were then lysed by shaking using Beadbug6 (4000 rpm, 30 s, 8 times). The lysates were then processed with the standard mitochondrial isolation protocol.

## Ovary immunofluorescence staining

Ovaries were dissected in Grace's media and then quickly fixed in 1xPBST supplemented with 4% formaldehyde for 15 min. Ovaries were then washed 3 times in 1x PBST supplemented with 0.1%BSA for 30 min per wash. Ovaries were then blocked in 1x PBST supplemented with 0.3% BSA + 5% horse serum for at least 1 h at room temperature. Primary antibodies were added: anti-Me31B (1:1,000 provided by Dr. Akira Nakamura, kumamoto University), anti-1B1(1:500), anti-VASA (1:500). The ovaries were incubated with primary antibodies overnight at 4 °C with gentle agitation. On the next day, the ovaries were washed 3 times for 1 h each wash in 1xPBST supplemented with 0.1% BSA + 0.5% horse serum. Secondary antibodies were then added and incubated for 2 h at room temperature with gentle agitation. Ovaries were then washed 3 times for 30 min, each wash in 1xPBST supplemented with +0.5% horse serum. Ovaries were stained with DAPI during the last wash step and mounted in Vectashield on slides.

The secondary antibodies (Alexa 488 and Alexa 568 from Molecular Probes/Invitrogen) were used at 1:1000. DNA was stained with 4,6diamidino-2-phenylindole (DAPI; Sigma) at a final concentration of 1 µg/ml.

## Western blotting analysis

Samples were prepared using $1 \times$ SDS-PAGE protein sample buffer (50 mM Tris-HCl (pH 6.8),1% SDS, 5% Glycerol, 1% β-mercaptoethanol, 0.002% bromophenol blue). After denature, the samples were loaded on 4–15% Tris-glycine gradient gels, and electrophoresis was run at 120 V for 1.5 hr. Gels were then transferred to PDVF membrane using Towbin's transfer buffer (25 mM Tris, 192 mM glycine, pH 8.3, 20% methanol) with a consant 250 mA for 2 h at 4 °C. The membranes were blocked for 1 h in 1xTBST with +3% nonfat milk. Primary antibodies were added as follows anti-K48 (1:1,000)(SCBT), anti-ATP5α (1:10,000)(ABCAM), anti-Actin (1:500)(sigma), anti- Flag (1:1,000), anti-Myc (1:1,000), anti-Tubulin (1:1,000)(DSHB# E7) and anti-laminB (1:1,000 DSHB). RPT2, RPT5, 20 S, and RPN12 antibodies were a gift from George Demartino's lab[46]. The blots were then incubated with primary antibody overnight at 4 °C with gentle shaking. On the next day, primary antibodies were removed, and the membranes were washed 3 times with $1 \times$ TBST. Appropriate secondary antibodies were added at a dilution of (1:10,000) and incubated for 1 h at room temperature with gentle agitation. Secondary antibodies were then removed, and the membranes washed 3 times with $1 \times$ TBST. Blots were then treated using the Pierce SuperSignal West Pico Chemiluminescent substrate (#34080) and developed by Bio-rad ChemiDoc™ MP imaging system.

## Native PAGE assay for assembled proteasome complex

Nondenaturing PAGE was conducted with NuPAGE™ 3 to 8% Tris-Acetate gradient gels (ThermoFisher, EA03785BOX). For analysis of lysate on native gels, staged oocytes were first homogenized with lysis buffer (50 mM Tris-HCl (pH 8.0), 5 mM MgCl2, 0.5 mM EDTA 100 µM ATP), and then centrifuged at 20000x$g$ for 15 min to remove cell debris. Protein levels were determined by Bio-Rad Bradford assay and adjusted to similar levels for the subsequent assays. ~50 µg adjusted cell lysates were mixed thoroughly and carefully with 5X sample loading buffer (250 mM Tris-HCl (pH 7.6), 50% glycerol, 60 ng/ml xylene cyanol).

680 Gel electrophoresis was performed with $1 \times$ Tris buffer with 100 µM ATP (No SDS). The whole process was performed at 4 °C. The gel electrophoresis started at 45 V for 0.5 h. The voltage was then raised to 90 V, and the gel was run for 3.5 h. Gels were then transferred to PDVF membrane using Towbin's transfer buffer (25 mM Tris base, 192 mM glycine) with a constant 25 mA for 18 h at 4 °C. Before transfer, the entire gel is soaked in transfer buffer (containing 1% SDS) for 15 min (Citation, I include this in email, George gave me this for reference. I follow the exact step and buffer for experiments). The remaining experimental procedures follow a normal western blotting assay. The primary antibodies to individual 26 S proteasome complex were added as follows: anti-20S (1:500), anti-Rpt2 (1:500), anti-Rpt5 (1:500), anti-Rpn12 (1:500) (George DeMartino). The blots were then incubated with primary antibody overnight at 4 °C with gentle shaking. On the next day, primary antibodies were removed, and the membranes were washed 3 times with 1xTBST. Appropriate secondary antibodies were added at a dilution of (1:10,000) and incubated for 1 h at room temperature with gentle agitation. Secondary antibodies were then removed, and the membranes washed 3 times with 1xTBST. Blots were then treated using the Pierce SuperSignal West Pico Chemiluminescent substrate (#34080) and developed by Bio-rad ChemiDocTM MP imaging system.

## Proteasome activity assays

Using methods based on ref. 47 with assistance from the George DeMartino's group we assayed proteasome activity using the following methods. Proteasome activity was measured by determining rates of enzymatic cleavage of 7-amino-4-methylcourmarin (AMC) from peptide substrates Suc-LLVY-AMC. For total proteasome activity, staged oocytes were homogenized with lysis buffer (20 mM Tris-HCl (pH 7.6), 20 mM NaCl, 1 mM 2-mercaptoethanol, 5 mM MgCl2, 100 µM ATP, and 0.1% NP-40) and then centrifuged at 20000x$g$ for 15 min to remove cell debris. Protein levels were determined by Bio-Rad Bradford assay and adjusted to similar levels for the subsequent assays. ~10 µg adjusted cell lysates (20 µl) were incubated with 150 µl substrate buffer (20 mM Tris-HCl, pH 7.6 @ 37 °C, 20 mM NaCl, 1 mM DTT, 100 µM ATP, 500 µM MgCl2). Measurements were carried out at 37 °C for 21 min in a Biotek FL600 fluorescence plate reader with filters at 380 nm excitation/460 nm emission. AMC fluorescence was monitored once 45 s during the assay, and progress curves were analyzed with kinetic software. Control groups include lysis buffer only and cell lysates group with 100 µM MG132. For mitochondria-associated proteasome activity, similar amounts of mitochondrial pellets were resolved in 100 µl mitochondrial isolation buffer and incubated on ice for 15 min with gentle pipetting every 3 min before assay. ~20 ul dissolved mitochondrial solution set aside for mitochondria quantification by probing ATP5α using western blotting. All mitochondria-associated

proteasome activity measurements were then normalized to mitochondria amount.

Our in-gel digestion assays were conducted as described in ref. 47, semiquantitative measures of 26 S proteasome activity were obtained by the overlay of AMC peptide substrates (No SDS in substrate buffer) in situ on proteins separated in freshly prepared native 4% polyacrylamide gels. After incubation at 37 °C for 20 min, AMC at the position of the protease in the gel responsible for its 26 S proteasome complex with either one or two regulatory caps was visualized by UV light.

## Metabolite measurements

For both triglyceride and glycogen measurements, females of the desired genotypes were dissected, and 200 oocytes of the corresponding stage were collected. The samples were then homogenized in 150 μl of 1 × PBST (0.1% triton) and heat–treated (5 min @100 °C). The resulting heat-treated lysate was then cleared by centrifugation at 13,000x$g$ for 3 min. Glycogen was then assayed using the glucose oxidase kit (Sigma, # GAGO20-1KT) and amyloglucosidase (Sigma, #1602) as described in ref. 15. Triglyceride levels were measured using triglyceride reagent (Sigma, # T2449) and free glycerol reagent (Sigma, # F6428) as described in ref. 15. Protein levels were assayed by using Bio–Rad protein assay reagent (# 5000006). All glycogen and triglyceride measurements were then normalized to the total protein level. All data presented is derived from at least 5 replicate samples, and each experiment was repeated at least 3 times.

## 3T3 cell model for quiescence

3T3 cells were cultured in DMEM ( + 10%FBS) until 90% confluent. The medium was removed, and cells were rinsed 2X in 1xPBS. Fresh starvation media was added (DMEM + 0.1% FBS), and cells were incubated for 3 days. Starvation media was then removed, and fresh DMEM( + 10%FBS) was added. After 2 days of reactivation, cells were then plated on a seahorse assay plate, and OCR and ECAR were measured.

## Seahorse analysis

The day prior to assay, each well is plated with 15,000 cells. On the day of the assay cell, culture media is removed and replaced with seahorse assay media supplemented with glutamine. Cells were first allowed to adapt to assay media for 2 h in the incubator. Once acclimated, the cells are assayed for basal OCR, ATP-dependent OCR, and maximal OCR using a standard mito-stress kit protocol. For the assay, we used 2 μM oligomycin and 1 uM FCCP for the injections. Once the assay is complete, cells are collected, and total protein levels are measured by Bradford protein assay to ensure the observed changes in respiration do not manifest from changes in cell number. To measure basal OCR of staged oocytes, 5 intact staged oocytes isolated from indicated genotypes were washed twice with Grace's insect media before loading into Seahorse assay plate. The oocyte samples were equilibrated in Grace's insect media at 25 °C for at least 30 min before measuring the basal OCR. All experiments were conducted on a Seahorse XFp machine.

## In vitro kinase assay

GSK-3 candidate targets are synthesized by TnT Quick coupled transcription/translation systems (Promega, # L1170). Two rounds of PCR reactions were used to generate cDNA templates with T7 promoter added in 5′ end and His-tag added in 3′ end. For the 1st round PCR reaction, forward primers with a part of T7 promoter sequence and reverse primers with His-tag sequence were used to generate intermediate PCR products. In the 2nd round PCR reaction, a common forward primer with the complete T7 promoter sequence and reverse primers were used to generate the final cDNA templates containing T7

promoter in 5′ end and His-tag in 3′ end for GSK3 candidate targets. After gel ~500 ng PCR template is added to an aliquot of the TnT Quick Master Mix and incubated in a 50 μl reaction volume for 90 min at 30 °C. The synthesized proteins are purified by NEBExpress® Ni Spin Columns (NEB, #1427 S) and then analyzed by SDS-PAGE. -20 ng synthesized candidate proteins were mixed with ~40 ng recombinant GSK-3 protein (Sino Biological #10044-H07B) in a total 25 μl reaction volume. The in vitro phosphorylation reaction was performed following ADP-Glo™ Kinase Assay (Promega, # V9101). (primers sequences located in Supplementary Data 5)

## GSK3-APEX2 labeling and enrichment of biotinylated GSK3-interacting proteins

Fly ovaries were dissected and gently dissociated in PBS. The ovarioles were incubated with 500 μM biotin-phenol in PBS for 30 min at room temperature with gentle agitation for every 5 min. After incubation, the substrate-containing solution was removed. To activate the APEX2 enzyme for protein labeling, 1 mM $H_2O_2$ in PBS was added to the samples for 1 min and then quickly quenched to minimize over labeling. To stop the labeling reaction, the samples were washed three times with PBS quenching buffer (1 mM sodium ascorbate, 2 mM Trolox, 5 mM sodium azide). Mitochondria were isolated from ~250 μg total cell lysate following the standard procedure. To enrich biotinylated proteins, mitochondria were first lysed in 500 μl RIPA buffer and then was mixed with 80 μL of streptavidin-coated magnetic bead slurry (Pierce # 88816) that was prewashed twice with RIPA buffer. The mixtures were incubated at room temperature for 1.5 h with rotation. After incubation, the beads were washed twice with 1 mL RIPA buffer, once with 1 mL of 2 M urea in 10 mM Tris·HCl (pH 8.0), and twice with 1 mL RIPA buffer. The biotinylated proteins were then mixed with 50 μL 1X SDS sample buffer and boiled at 100 °C for 10 min to release biotinylated proteins. The samples were then loaded on 4–15% Trisglycine gradient gels, and electrophoresis was stopped when samples run -1 cm from well. The gel slices were cut and sent for mass spectrum analysis.

## Proteomics

Ovaries were dissected from mature adult female flies in Grace's insect media, and the individual follicles were separated with forceps. Samples were then collected at 300 oocytes/sample. Follicle cells were removed from stage 14 oocyte as described above. Mitochondria were isolated as described above, and protein was precipitated via TCA. Mitochondrial protein was constituted in 500 mM TEAB( + 1%SDS), and 100 ug was taken for processing and analysis.

Proteins in solution (100 ug in 50 ul) were reduced with DTT (x3) and then alkylated with iodoacetomide (x3). 15 ul (30 ug) of the protein solution was precipitated the remainder was frozen. The TCA/acetone pellet was reconstituted with 20ul 500mmTEAB(+10 uL water) and sonicated for 10 min. The sonicated solution was then proteolyzed with trypsin (frozen, Promega) 20 ng/ul. Following digestion, peptides were desalted on Oasis u-HLB plates (Waters). The digested peptides were reconstituted in 8 ul (2%ACN/0.1%FA). 1 ul (12% of the sample) was analyzed by LC/MSMS on a nano- LC ESI LTQ_Orbitrap_Velos (ThermoFisher) in FTFT using 90 min total gradient, resolution at 400 Da 60 K for Full MS and 15 K for MS2. The injected peptides showed an abundant (~1 uG) base peak chromatograph.

## LC/MS analysis

Samples are analyzed by targeted LC/MS metabolomics with the assistance of the UT Southwestern metabolomics facility (https://cri.utsw.edu/facilities/metabolomics-facility/). Using Sciex SIMCA software, we will analyze these data sets and perform Partial Least Square analysis to examine sample clustering. From the PLS analysis, we will

identify metabolites that exhibit a high VIP score (>1.0) as candidate compounds that contribute to the reprogramming of progeny metabolism.

## GC/MS metabolomics analysis

GC/MS metabolomics analysis was conducted exactly as described in (ref. 15). Samples were then collected at 250 oocytes/sample and flash-frozen in liquid nitrogen, and stored at −80 °C until processed. Vials containing oocytes were removed from the −80 °C freezer, and the contents were immediately transferred into a bead mill tube containing 1.4 mm ceramic beads (MoBio Laboratories, Carlsbad, CA). This tube contained 400 μL of 80% MeOH that had been chilled to −20 °C. Whole flies were homogenized for 30 s at 6.5 m/sec with an Omni Bead Ruptor 24 bead mill (Omni-Inc, Kennesaw, GA). Cell debris was removed by centrifugation at $14,000 \times g$ for 5 min at 4 °C. The supernatant was transferred to microcentrifuge tubes, and the solvent was removed *en vacuo*.

GC-MS analysis was performed with a Waters GCT Premier mass spectrometer fitted with an Agilent 6890 gas chromatograph and a Gerstel MPS2 autosampler. Dried samples were resuspended in 40 μl of a 40 mg/mL O-methoxyamine hydrochloride (MOX) in pyridine and incubated for one hour at 30 °C. 25 μl of this solution was added to each autosampler vial. Ten microliters of N-methyl-N-trimethylsilyltrifluoracetamide (MSTFA) was added automatically via the autosampler and incubated for 60 min at 37 °C with shaking. After incubation, 3 μl of a fatty acid methyl ester standard solution was added via the autosampler. 1 μl of the prepared sample was injected into the gas chromatograph inlet in the split mode at a 10:1 split ratio with the inlet temperature held at 250 °C. The gas chromatograph had an initial temperature of 95 °C for one minute, followed by a 40 °C/min ramp to 110 °C and a hold time of 2 min. This was followed by a second 5 °C/min ramp to 250 °C, the third ramp to 350 °C, then a final hold time of 3 min. A 30 m Phenomex ZB5-5 MSI column with a 5 m long guard column was employed for chromatographic separation.

Data was collected using MassLynx 4.1 software (Waters). A two-step process was employed for data analysis, a targeted followed by non-targeted analysis. For the targeted approach, known metabolites were identified, and their peak area was recorded using QuanLynx. This data was transferred to an Excel spreadsheet (Microsoft, Redmond WA). Metabolite identity was established using a combination of an in-house metabolite profile library developed using pure purchased standards and the commercially available NIST library. When reporting each metabolite, those with absolute identity are not qualified; those that are identified using a NIST library are noted using a percentage of certainty produced by the NIST software. Not all metabolites are observed using GC-MS. This is due to several reasons, one being that they present at very low concentrations. A second is they are not amenable for the GC-MS procedure due to either being too large, too volatilize, are a quaternary amine such as carnitine, or do not properly ionize. Metabolites that do not ionize well include oxaloacetate and arginine. Cysteine was observed depending upon cellular conditions, it often forms disulfide bonds with proteins and is at low intracellular concentration. Statistical analysis was performed using MetaboAnalyst Ver. 2. data was converted to a.csv file and uploaded into the MetaboAnalyst server. Data were mean filtered to remove variables of very small values, normalized by autoscaling, and fold change, *t*-test, and volcano plot were generated.

The data is compiled from 10 samples/stage.

## Quantification and statistical analysis

All metabolite measurements are conducted on at least 5 biological replicate samples per experiment, and each experiment is replicated at least 3 times. All proteasome activity assays represent 3 independent experiments on independent samples data points represent the average of each experiment unless stated other wise. All error bars represent the standard deviation of the sample group. Significance values are calculated by student's t-test for pairwise comparisons and One-way ANOVA for experiments where more than 2 sample groups are compared. All statistical analysis is conducted on independent biological replicates.

## Graphic software

All graphic models presented in this manuscript were made using Biorender (license# *FH241GMNIX*).

## Reporting summary

Further information on research design is available in the Nature Research Reporting Summary linked to this article.

## Data availability

Our proteomic data sets are available on line at massive.ucsd.edu under the accession numbers MSV000087642 and MSV00008764. These data sets will be publicly available upon publication. Raw data for all figures is provided in source data. All other data will be made readily available upon request. Source data are provided with this paper.

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

## Acknowledgements
We want to thank B.H.Graham and the BDSC for sharing vital stocks and reagents. We would also Like the thank M. Buszczak, D.Pan, and H. Hocaoglu for helpful comments and suggestions during this manuscript's preparation. M.H.S. supported by generous funding from the NIH/NIA (R01AG067604), the Welch Foundation (I-2015-20190330), the W.W. Caruth Jr foundation, and the UTSW Endowed Scholars program. Y. L. is supported by the National Institutes of Health (R35GM118118) and the Welch Foundation (I-1560). G.N.D. is supported by the National Institutes of Health (R01GM129088)

## Author contributions
S.Y., F.Z., L.W., and M.H.S. conducted the experiments and analyzed the data described in this manuscript. S.Y., G.N.D., Y.L., and M.H.S. designed the experiments and provided reagents. S.Y. and M.H.S. wrote the manuscript.

## Competing interests
The authors declare no competing interests.

## Additional information

 13

