## [Peer Review File · Nature Communications]

Highly conserved shifts in ubiquitin-proteasome system (UPS) activity drive mitochondrial remodeling during quiescenceREVIEWER COMMENTS

Reviewer #1 (Remarks to the Author):

Review on the manuscript “Highly conserved shifts in ubiquitin-1 proteasome system (UPS) activity drive mitochondrial remodeling during quiescence” by Yue et al.

In this study, the authors use the model system of quiescent oocytes from *Drosophila* to show that mitochondrial respiratory quiescence (MRQ) is associated with an increased recruitment of 26S proteasomes to mitochondria. Their data suggest that GSK3, a crucial regulator of MRQ, is regulating phosphorylation of mitochondria associated proteins such VDAC which has an effect on proteasome recruitment and activity. The authors also include data on starvation induced quiescence of 3T3 mammalian fibroblasts and developmental quiescence of the fungus *Neurospora crassa* to provide evidence for evolutionary conservation of this process.

The finding of 26S proteasome recruitment and activation to mitochondria as a conserved feature of quiescence in different species, is novel and highly interesting. However, the data lack important controls and the conclusions are not always supported by the evidence. Moreover, the authors do not properly include the current state of knowledge on the interplay between proteasome and mitochondria, which is already well established, and thus overinterpret their findings.

Specific comments:

1. Controls for effective silencing of single proteins are lacking throughout the whole manuscript and need to be provided.
2. The reproducibility of the findings is not almost demonstrated and needs to be carefully shown using multiple independent experiments and samples and include statistical analysis. Please provide that for all experiments and clearly state the number of independent experiments, normalization of the data and statistical tests applied.
3. The quality of the mitochondrial isolations is not properly shown. Proper controls need to be included which demonstrate the absence of other organelle fractions e.g. the ER. It also needs to be controlled for the potentially altered mitochondrial structure in MRQ, which might result in an altered subcellular localization which could explain the altered 26S proteasome activity.
4. The alteration in 26S proteasome activity needs to be better controlled: please show activity assays for all active sites. Please provide the full native gel, including 20S and 26S complexes as well as full blots for the different proteins (see e.g. STAR Protoc. 2021 May 4;2(2):100526). Please show reproducibility of

activity assays using independent oocyte preparations and statistical analysis. Same applies to Western blot. What is the 20S band in Ex Fig 1B? There should be multiple bands for the 20S when using a pan-antibody. Only 20S and Rpt2 were elevated but not Rpt5 and Rpn12 arguing against concerted upregulation of 19S and 20S subunits. Please provide RNA data to discriminate between transcriptional and assembly or recruitment effects.

5. Provide controls for intact proteasome complexes. The proteasome activity assay buffer does not contain ATP. Accordingly, the 26S will most probably fall apart.

6. Treatment of oocytes with 50 μ M MG132 for 2 hours will alter mitochondrial function and is not a low dose proteasome inhibitor treatment as also shown by the authors in Ext Figure 1A. Alterations in mitochondrial function upon proteasome inhibition is a well-known phenomenon (e.g. first description: Sullivan et al., J. Biol. Chem 2004 May 14;279(20):20699-707) and does not prove that recruitment of proteasomes to the mitochondria drives mitochondrial remodeling.

7. The figure legends lack any information on the number of independent experiments, statistics details on the experiments and need to be significantly improved. Please clearly show where technical or biological replicates were used.

8. The study lacks mechanistical data on how the proteasome is recruited to the mitochondria. It appears to involve VDAC, but it is not clear how that might work.

9. Conclusions are not always supported by the data. e.g. Figure 2: Reduced OCR via inhibition of FAO, activates proteasome activity associated with mitochondria and K48 ubiquitination. Vice versa, activation of OCR by inhibition of GSK inhibited proteasome activity. These data show the OCR activity and proteasome activity are coupled and not “These data indicate that GSK induces mitochondria-associated proteasome activity during quiescence, in part, through inhibition of the FAO pathway.” (line 131ff). Similar in line 212ff: “VDAC phosphorylation by GSK3 triggers the recruitment of the proteasome to the mitochondria and promotes the turnover of mitochondrial outer membrane proteins.” This conclusion is not supported by the data which merely show that VDAC is required for increasing proteasome activity associated with mitochondria when this is activated by silencing FAO genes. The data do not show that phosphorylation by GSK is crucial in that process. GSK3 is able to phosphorylate VDAC but whether this is required for its effect on proteasome activity is not shown. Again in line 253ff: “activity is significantly lower and proteasome recruitment to the mitochondria further compromises cytosolic protein turnover rate.” The authors merely show distinct proteasome activities in total versus mitochondrial fraction. One cannot deduce causality from that observation. Moreover, as data are normalized to controls, activities should be assayed side by side and normalized to total activity to estimate how much the mitochondria-associated proteasome activity contributes to total activity. This approach should also be done also for native page-assays.

10. The authors do not investigate what functional consequence the 26S recruitment to mitochondria has. This is a weakness of the study.

11. Figure 3: What happens to proteasome activity and recruitment to mitochondria in GSK silenced/VDAC het cells in a similar experiment as Figure 3J.

12. Discussion is far-fetched and does not include any published data on the interplay of mitochondria and proteasome. It is well established that the proteasome is localized to mitochondria. The interplay

between mitochondrial respiratory chain function and proteasome activity is also well established but not mentioned by the authors.

Reviewer #2 (Remarks to the Author):

Summary

The proper regulation of quiescence is critical for many biological processes, including tissue homeostasis and oocyte development. Previous work by the senior author established *Drosophila* oogenesis as a system to study mitochondrial respiratory quiescence (MRQ), showing that suppression of insulin signaling/activation of GSK3 triggers MRQ and glycogen accumulation in the mature oocyte. In this study, Yue et al. investigate the molecular mechanisms linking GSK3 activation and MRQ in *Drosophila* oocytes, and their evolutionary conservation in mammalian cells and fungi using a combination of genetics, cell biology, cell fractionation, biochemistry, proteomics and metabolomics approaches. The authors show that the expression and activity of the ubiquitin-proteasome system (UPS) and K48 ubiquitination (followed by turnover of ubiquitinated proteins) increases as oocytes enter quiescence, and the levels of proteins involved in fatty acid oxidation are reduced. Conversely, knockdown of GSK3 in oocytes results in elevated mitochondrial activity and increased levels of proteins involved in fatty acid oxidation. Consistent with their model that GSK3 inhibits fatty acid oxidation and trigger proteasome recruitment to mitochondria, knockdown of mitochondrial fatty acid oxidation genes led to increased levels of mitochondria-associated proteasome activity and increased K48 ubiquitination. Using proximity labeling combined with cell fractionation, the authors identified GSK3-associated proteins and showed that VDAC (a mitochondria porin ion channel in the outer mitochondrial membrane), and components of the TOM complex (a protein complex in the outer mitochondrial membrane involved in protein translocation into mitochondria) are directly phosphorylated by GSK3 in vitro. They also showed that overexpression of GSK3 (but not a kinase dead mutant) in 293T mammalian cells leads to very reduced levels of Tom22 and VDAC, and they also identified the specific phosphorylation site in VDAC required for this effect. They propose the model that GSK3 recruits the proteasome to the mitochondria by directly phosphorylating Tom22 and VDAC at the outer mitochondrial membrane and by repressing fatty acid oxidation in a VDAC-dependent manner. Remarkably, the authors show similarities between developmental quiescence in *Drosophila* oocytes and other types of induced or developmental quiescence. They show that amino acid deprivation induces precocious MRQ in developing *Drosophila* follicles (based on TMRE staining). They also show that serum deprivation-induced quiescence in mouse fibroblasts leads to reduced mitochondrial activity (and of mitochondrial proteins), increased glycogen levels, and proteasome recruitment to the mitochondria, and that developmentally quiescent *Neurospora crassa* spores also have elevated total and mitochondria-associated proteasome activity, demonstrating the evolutionary conservation of MRQ mechanisms.

Critique

The findings reported in this study represent a significant advance in our understanding of highly conserved mechanisms underlying quiescence, and should be of interest to a wide range of scientists. Notably, the authors combine a variety of powerful experimental approaches and distinct experimental systems to address important questions, test novel hypotheses, and open new directions of thought in the field. However, there are some concerns that should be addressed by the authors regarding methodological details, clarification for some of their data/conclusions, and the need to soften a couple of their conclusions, as outlined below.

Specific points:

1) Throughout the methods, more information is needed to demonstrate rigor and reproducibility. For every single biochemical, genetic, microscopy, etc, experiment, the authors should clearly specify how many times the experiments were repeated, sample sizes, and statistical methods used to analyze the data. More details are also needed for some of the experiments to ensure sufficient information for reproducibility. For example, in line 587, it is not clear what “sufficient” means precisely. Under “TMRE staining”, clarify if experiment with MG132 has insulin added to PBS as well (lines 599-600). For all construct cloning, include name of cDNAs used and where they were obtained, provide all primer sequences in a table, explain how mutations were introduced, etc. For mitochondrial isolation/native PAGE assay, explain number of staged oocytes used (line 625) and how oocytes were homogenized – motorized pestle? FastPrep machine? (line 627). For western blotting analysis, include what buffer the samples were in prior to loading onto denaturing versus native gels, where gradient gels were purchased, composition of transfer buffers and of TBST, specify secondary antibodies and source (also for primary antibodies), etc. Similarly, include more details for Seahorse assays, in vitro kinase assays (including vectors for targets), generation of APEX constructs and fly lines, etc.

2) Extended data Fig. 2C legend: authors should explain more precisely what criteria were used for the classifications of normal, moderate and severe.

3) Extended data Fig. 3A: the authors did not include the expression data for the APEX fused to CD8 control.

4) There is a general concern that all RNAi experiments were done using single RNAi lines.

5) In Fig. 3E (and lines 176-177), the overexpression of the kinase dead GSK3 is sufficient to reduce the levels of Tom22 and VDAC (albeit to a much lesser extent than the wild type GSK3) in 293T cells. The authors should briefly discuss possible reasons for that. (e.g. Is it possible that GSK3 has some kinase-independent roles?)

6) The authors should be extra clear in their explanation of a seemingly confusing but interesting aspect of their model/data: VDAC phosphorylation by GSK3 appears to trigger proteasome recruitment to the mitochondria and subsequent degradation of VDAC in 293T cells (see lines 168-183). Yet, knockdown of VDAC has the same phenotype as GSK3 knockdown (which should lead to MORE stable VDAC) in *Drosophila* oocytes (Fig. 3H, I), and VDAC loss of function dominantly suppresses the increased proteasome activity induced by knockdown of fatty acid oxidation genes (Fig. 3J). These data seem to imply that VDAC plays an important role in inducing MRQ prior to its degradation by the proteasome. [Although VDAC knockdown paradoxically reduces oxygen consumption and increases glycogen levels at the same time that it increases TMRE levels (Extended data Fig. 6A-C) and reduces mitochondria-associated proteasome activity (Fig. 3I)!?] A clarification of these confusing points would be very helpful to the readers.

7) Lines 227-233: in these sentences, the author should soften their conclusions regarding the mechanisms involved in amino acid deprivation-induced quiescence because they did not directly investigate proteasome recruitment to mitochondria in this system. The tone of their conclusions should be adjusted accordingly.

8) The conclusion in lines 261-263 is similarly overstated.

9) Regarding some key aspects of their model, the authors should consider testing whether overexpression of key fatty acid oxidation enzymes in the *Drosophila* germline prevents mature oocyte MRQ (and recruitment of the proteasome to mitochondria). They should also speculate/discuss how they envision the proteasome recruitment to the mitochondria in their model: does it associate with the outside of mitochondria, or does it get imported somehow into the mitochondria? They should also draw a connection of their study with published work on the import of proteins into mitochondria and whether mitochondria import might be necessary for proteasome recruitment to mitochondria.

10) The authors should consider subdividing their manuscript into different sections with subtitles to improve the flow and clarity of the manuscript.

Typos, etc:

- Line 18: M.H.S. instead of M.S.?

- Lines 33-34: sentence should be re-written to make more sense grammatically and otherwise.

- Line 136: incorrect figure citation

- Line 187: authors should clarify more precisely what they mean by this sentence and provide a reference citation.
- Line 198: reference citation missing for FAO-related metabolites
- Line 204: data “show”
- Lines 218-219: clarify in which organisms/systems this has been previously shown.
- Line 318: typos
- All figure legends are missing a figure title, and the figure legends are somewhat cursorily written.
- In figure 2, the order of panels does not correspond to the figure legend.
- In figure 4G legend, “experimental strategy” does not seem appropriate because they are just showing the “lifecyle” of *N. crassa*.
- Extended data Fig. 4D: why not include Mut1 as a comparison as well?
- Line 582: small case for *crassa*
- Line 650: 1B1 (not anti-1B1)
- Line 664: instead of “gels” were transferred, “proteins” were transferred.
- Line 666: A “:” is missing after “follows”
- Lines 682-683: instead of “the rest experimental processes”, “the remaining experimental procedures”
- Edit for additional typos.

Reviewer #3 (Remarks to the Author):

Review of: Highly conserved shifts in ubiquitin-proteasome system (UPS) 2 activity drive mitochondrial remodeling during quiescence 3 4 Authors: Sibiao Yue1, George N. DeMartino1, FangZhou Zhao1, Yi Liu1 5 , Matthew H. Sieber1*

This is an excellent paper concerning the cell quiescence, a state found in germ cells and other cells that are deeply prevented from proliferation and differentiation. The authors chose an excellent system the fly ovariole, which houses oocytes that will stored in a stable state until hormonally stimulated to

mature and be ovulated. The transition to that state is an important problem and likely to be general and the *Drosophila* ovary is a great system. In the quiescent state the mitochondria are also quiescent in a state known as mitochondrial respiratory quiescence MRQ and these authors see this as primary. They provide excellent proteomic (MS) and other evidence that proteasome activity associated with the mitochondria is more active. They implicate GSK3 activity in phosphorylating outer membrane proteins and particularly VDAC an anion transporter that brings in much of the metabolic intermediates. They show that this is a key step driving quiescence and hormonally sensitive. What is truly fascinating here is that energy metabolism is regulated by proteolysis and this in turn is regulated by proteasome localization. If there is one quibble is that the dynamics should be visualizable at the light microscope level. They argue that there is depletion of the proteasome from the cytosol. Microscopic monitoring would allow the kinetics to be better determined, to note if localization is spatial on a single mitochondrion or whether it only affects particular mitochondria. It would be possible to do a better dose response. This could be a static experiment with anti proteasome antibodies or perhaps a dynamic experiment with fluorescent markers. I do not think this is essential as there is convincing enough data on mitochondrial state.

I find little to complain about here and at a high level I am sure that this will be important. It is an unusual problem, it is an important problem. It is mechanistically cleanly presented. The tests of necessity and sufficiency are clear. Of course there may be several ways of initiating MRQ and several different outcomes. The authors followed the VDAC channel but it is not clear if it is the only path even though it looks necessary.

Reviewer #4 (Remarks to the Author):

This manuscript describes the authors' studies investigating the role of the ubiquitin-proteasome system in driving mitochondrial remodeling during quiescence. The authors argue that an elevated number of proteasomes in quiescent cells and their increased recruitment to mitochondria results in a phenomenon termed mitochondrial respiratory quiescence. More specifically, GSK3 was found to trigger proteasome recruitment to mitochondria. GSK3, through outer membrane proteins including VDAC, affects proteasome degradation. These findings were reported in *Drosophila*, mouse and fungi.

Comments

The authors may have elucidated a pathway involving fatty acid oxidation affecting proteasome activity through VDAC. There are some data that suggest that this pathway may exist. Fig 3J does suggest that VDAC may be involved in a pathway in which fatty acid oxidation affects proteasome activity. However,

there are many concerns about the data and conclusions at this point. The authors are trying to establish a very complex model. At every point, concerns about rigor make it difficult to agree with the authors conclusions. Many important controls and information needed to agree with the authors conclusions are missing. The authors are trying to make an argument about a pathway that is active in quiescence ,but most of the figures do not show proliferating and quiescent cells so the selectivity for quiescence is not clear. More rescue experiments would be needed to demonstrate causality in the signaling pathway the authors hypothesize. For instance, there is no evidence that VDAC phosphorylation, rather than simply VDAC levels, is involved in this pathway. Further, the work is almost exclusively performed in *Drosophila*, but the authors seek to make very broad claims about this pathway being consistent in all quiescent cells that are not justified.

The reduced mitochondrial oxidation of fatty acids that the authors argue is a hallmark of quiescence is not consistent with the literature. For instance, in Ito et al (Nat Med 2012), the authors found that inhibiting fatty acid oxidation in hematopoietic stem cells resulted in exit from quiescence. The authors should review this literature further and frame their manuscript accordingly.

The authors need to explain the *Drosophila* oocyte model: what sample are being taken and when and what is considered quiescent and why?

Fig 1b: The figure legend should explain what is being shown here. How many independent samples from how many flies were used to generate these data? How many times was this performed? What does St1-8, St14, 0-2h and 16-20h refer to?

In general, this information should be added to all of the figure legends.

Line 72: by what criteria were these genes “significant” in the RNA seq data? What samples are being compared to define significance?

Line 77: what does “onset of quiescence” mean? How was this modeled in *Drosophila*?

What are we comparing in 1D? Is there quantification on multiple samples that would convince us that these two lanes are different?

Fig 1E: what controls were performed to confirm that mitochondria were isolated for these experiments?

Where are the data to support this statement: "Moreover, we observed that K48 ubiquitination returns to normal in the hours after the onset of quiescence, suggesting that these ubiquitinated proteins have been turned over."?

Are these proteasomes inside the mitochondria or attached to them on the outside?

1G: Are these mitochondrial proteins being ubiquitinated? Can the authors show any examples?

Fig 2B: GSK3 inhibition affects oocytes as expected from the literature, but these results don't make it clear that there is a quiescence-specific effect

Fig 2B and 2C: Where is the validation of GSK3 RNAi knockdown? It would be best to have more than 1 siRNA to ensure the results aren't off target effects. This is true for all of the RNAi experiments.

Fig 2H and 2I: are these performed in quiescent cells? How do quiescent and proliferating cells compare for these assays?

Extended data Fig 3: It's clear that mitochondrial proteins were pulled down, but it is not clear how many non-mitochondrial proteins were pulled down. What fraction of the proteins are mitochondrial?

Fig 3G: what are we looking at in this figure? What is the difference between lanes 1 and 2? Lanes 3 and 4? Are they two examples of the same thing? Have these data been quantified? Is there a statistically significant difference?

Fig 3J: why use heterozygous VDAC lines?

These knockout and knockdown models do not demonstrate that VDAC phosphorylation, as shown in the schematic, is important for proteasome activity.

Fig 4 If the authors would like to make the argument that cytosolic proteasome-mediated protein degradation is being compromised during quiescence, it is suggested that they use a better model for monitoring cytoplasmic proteasome activity. It is also suggested that they use additional measures for the presence of protein aggregates in addition to Me31B. It is not possible for the reader to agree that there is a systemic effect on protein degradation from these data alone.

All figures: why is it that when cells have higher levels of proteasome activity, K48 ubiquitination is higher?

We thank the reviewers for their comments and their input. It has greatly improved the clarity of the manuscript. We have attempted to examine all of the reviewer comments in multiple systems and using multiple methodologies. We have provided additional activity data for our proteasome studies that strengthen and support our original findings. We have also provided more ubiquitination and Co-IP data that provide more insight into the role of VDAC in proteasome recruitment. To support our data that examines the role of proteasome recruitment in cytosolic proteostasis we have conducted new experiments that directly measure cytosolic proteasome activity.

Reviewer#1

Original comments

Review on the manuscript "Highly conserved shifts in ubiquitin-1 proteasome system (UPS) activity drive mitochondrial remodeling during quiescence" by Yue et al.

In this study, the authors use the model system of quiescent oocytes from Drosophila to show that mitochondrial respiratory quiescence (MRQ) is associated with an increased recruitment of 26S proteasomes to mitochondria. Their data suggest that GSK3, a crucial regulator of MRQ, is regulating phosphorylation of mitochondria associated proteins such VDAC which has an effect on proteasome recruitment and activity. The authors also include data on starvation induced quiescence of 3T3 mammalian fibroblasts and developmental quiescence of the fungus Neurospora crassa to provide evidence for evolutionary conservation of this process.

The finding of 26S proteasome recruitment and activation to mitochondria as a conserved feature of quiescence in different species, is novel and highly interesting. However, the data lack important controls and the conclusions are not always supported by the evidence. Moreover, the authors do not properly include the current state of knowledge on the interplay between proteasome and mitochondria, which is already well established, and thus overinterpret their findings.

Specific comments:

- 1. Controls for effective silencing of single proteins are lacking throughout the whole manuscript and need to be provided.*
- 2. The reproducibility of the findings is not almost demonstrated and needs to be carefully shown using multiple independent experiments and samples and include statistical analysis. Please provide that for all experiments and clearly state the number of independent experiments, normalization of the data and statistical tests applied.*
- 3. The quality of the mitochondrial isolations is not properly shown. Proper controls need to be included which demonstrate the absence of other organelle fractions e.g. the ER. It also needs to be controlled for the potentially altered mitochondrial structure in MRQ, which might result in an altered subcellular localization which could explain the altered 26S proteasome activity.*
- 4. The alteration in 26S proteasome activity needs to be better controlled: please show activity assays for all active sites. Please provide the full native gel, including 20S and 26S complexes as well as full blots for the different proteins (see e.g. STAR Protoc. 2021 May 4;2(2):100526). Please show reproducibility of activity assays using independent oocyte preparations and statistical analysis. Same applies to Western blot. What is the 20S band in Ex Fig 1B? There should be multiple bands for the 20S when using a pan-antibody. Only 20S*

and Rpt2 were elevated but not Rpt5 and Rpn12 arguing against concerted upregulation of 19S and 20S subunits. Please provide RNA data to discriminate between transcriptional and assembly or recruitment effects.

5. Provide controls for intact proteasome complexes. The proteasome activity assay buffer does not contain ATP. Accordingly, the 26S will most probably fall apart.

6. Treatment of oocytes with 50 μ M MG132 for 2 hours will alter mitochondrial function and is not a low dose proteasome inhibitor treatment as also shown by the authors in Ext Figure 1A. Alterations in mitochondrial function upon proteasome inhibition is a well-known phenomenon (e.g. first description: Sullivan et al., J. Biol. Chem 2004 May 14;279(20):20699-707) and does not prove that recruitment of proteasomes to the mitochondria drives mitochondrial remodeling.

7. The figure legends lack any information on the number of independent experiments, statistics details on the experiments and need to be significantly improved. Please clearly show where technical or biological replicates were used.

8. The study lacks mechanistical data on how the proteasome is recruited to the mitochondria. It appears to involve VDAC, but it is not clear how that might work.

9. Conclusions are not always supported by the data. e.g. Figure 2: Reduced OCR via inhibition of FAO, activates proteasome activity associated with mitochondria and K48 ubiquitination. Vice versa, activation of OCR by inhibition of GSK inhibited proteasome activity. These data show the OCR activity and proteasome activity are coupled and not "These data indicate that GSK induces mitochondria-associated proteasome activity during quiescence, in part, through inhibition of the FAO pathway." (line 131ff). Similar in line 212ff: "VDAC phosphorylation by GSK3 triggers the recruitment of the proteasome to the mitochondria and promotes the turnover of mitochondrial outer membrane proteins." This conclusion is not supported by the data which merely show that VDAC is required for increasing proteasome activity associated with mitochondria when this is activated by silencing FAO genes. The data do not show that phosphorylation by GSK is crucial in that process. GSK3 is able to phosphorylate VDAC but whether this is required for its effect on proteasome activity is not shown. Again in line 253ff: "activity is significantly lower and proteasome recruitment to the mitochondria further compromises cytosolic protein turnover rate." The authors merely show distinct proteasome activities in total versus mitochondrial fraction. One cannot deduce causality from that observation. Moreover, as data are normalized to controls, activities should be assayed side by side and normalized to total activity to estimate how much the mitochondria-associated proteasome activity contributes to total activity. This approach should also be done also for native page-assays.

10. The authors do not investigate what functional consequence the 26S recruitment to mitochondria has. This is a weakness of the study.

11. Figure 3: What happens to proteasome activity and recruitment to mitochondria in GSK silenced/VDAC het cells in a similar experiment as Figure 3J.

12. Discussion is far-fetched and does not include any published data on the interplay of mitochondria and proteasome. It is well established that the proteasome is localized to mitochondria. The interplay between mitochondrial respiratory chain function and proteasome activity is also well established but not mentioned by the authors."

We thank reviewer#1 for their comments. Their input has greatly aided to the clarity of the manuscript and strengthened our findings. We have added more detail regarding experimental number and now provide source data for every figure and extended data figure.

Response to specific comments:

“Controls for effective silencing of single proteins are lacking throughout the whole manuscript and need to be provided”

- We apologize for omitting these data. We now include q-PCR data that shows the efficiency of the RNAi transgenes used in the study.

“The reproducibility of the findings is not almost demonstrated and needs to be carefully shown using multiple independent experiments and samples and include statistical analysis. Please provide that for all experiments and clearly state the number of independent experiments, normalization of the data and statistical tests applied.”

We thank the reviewer for pointing this out and have expanded the description of each experiment in the figure legends and in the methods section. All data provided are the result of multiple independent experiments and all statistics are calculated using independent biological replicates. Precise details for each experiment can now be found in the figure legends and the methods section. We now also provide source data for every figure and extended data figure.

“The quality of the mitochondrial isolations is not shown. Proper controls need to be included which demonstrate the absence of other organelle fractions e.g. the ER.”

- The purity of our mitochondrial fractions is depicted in Extended data Fig 2. As the reviewer suggested we attempted to examine ER content with markers (KDEL, and calreticulin) and found the western blot and IHC signal to be very low in mature quiescent eggs similar to what has been observed in the literature (Lighthouse et al 2008). This likely stems from the fact that during *Drosophila* oogenesis the ER is broken down and reorganized during vitellogenesis (Lee and Cooley JBC 2007). Moreover, during nurse cell breakdown (the beginning of quiescence) autophagy and caspases are activated and many organelles, such as the ER and nurse cell nuclei, are broken down and trafficked into oocyte to provide material to support growth during embryogenesis (Work from Kim McCall's lab at Boston University). This ER breakdown likely supports the 95% reduction in global translation and disassembly of polysomes observed in quiescent eggs (Lovett and Goldstein 1977, Kronja et al 2015 Cell Reports). Ultra-structural studies show the ER is very fragmented and sparse in quiescent eggs. In fact the fragmented ER can only be easily detected when stained for catalase activity in these studies (Giorgi and Deri 1976). Similar changes in ER have been observed in mouse oocytes where ER levels are very low during quiescence (GV stage). However, ER levels increase and form a perinuclear network as eggs are activated during GVBD (FitzHarris et al Dev. Biology 2007). The exception to this organelle breakdown is the mitochondria which increase 3-4 fold in number and are actively transported into the oocyte during oogenesis in the balbiani body and through the ring canals.

“The alteration in 26S proteasome activity needs to be better controlled: please show activity assays for all active sites. Please provide the full native gel, including 20S and 26S complexes as

well as full blots for the different proteins (see e.g. STAR Protoc. 2021 May 4;2(2):100526). Please show reproducibility of activity assays using independent oocyte preparations and statistical analysis. Same applies to Western blot. What is the 20S band in Ex Fig 1B? There should be multiple bands for the 20S when using a pan-antibody. Only 20S and Rpt2 were elevated but not Rpt5 and Rpn12 arguing against concerted upregulation of 19S and 20S subunits. Please provide RNA data to discriminate between transcriptional and assembly or recruitment effects.”

- We appreciate the reviewer’s input and now provide additional data using a proteasome activity probe (Ubiq-018; Me4BodipyFL-Ahx3Leu3VS) that monitors the activity of all three enzymatic activities of the proteasome. These data (shown in extended data figure 1) demonstrate an increase in each of the proteasome’s three catalytic activities. We also provide full western blots for our native PAGE analysis in source data. The protocol cited by the reviewer cannot be used to address this question due to the presence of detergent (digitonin in the protocol) which would solubilize the mitochondrial membranes and make the studying this process impossible. This likely also contribute to the low levels of 20S in our experiments. However, the relative content and distribution of proteasome complexes of 26S holoenzymes and free 20S core particle are known to vary in different cell and tissue types and in different physiologic states. Here, in *Drosophila* oocytes under the conditions studied, we observe that the majority of the proteasome is in the form of double-capped 26S holoenzyme, with little to no uncapped 20S core particle. All quantitative data provided regarding proteasome activity assays reflect multiple biological replicates from multiple experiments. Because we hand-dissect staged oocytes for these studies, the amount of mitochondrial purified is severely limited. Accordingly, every data point in these graphs represents an independent set of oocytes. None of our data reflects technical replicates but, instead, are completely independent biological replicates. Similarly, our zymography and western blot data also represent independent samples, thereby strongly supporting the observations from our activity assays. A more detailed description of the number of biological replicates used in this study has been added to the figure legends. Moreover we have overlaid individual data points onto all of our graphs throughout the manuscript.
- Regarding RNA data requested by reviewer#1 we refer you to our original submission text and Extended data table 1:

“Using our previously published RNA-Seq datasets that examine the changes in gene expression that occur as oocytes enter cellular quiescence¹⁶, we observed a significant 1.4-2.5 fold increase in the mRNA expression of 26 genes involved with the proteasome and the UPS systems. These genes includes 20S core subunit factors and genes involved with the 19S regulatory cap (Extended data Table 1), suggesting that this increase in proteasome activity is driven by 26S biosynthesis.”

These data can be found Extended Data Table 1. We apologize if this was not clear in our previous submission and we have stated this more directly in the in the current resubmission.

Here is revised version of the text:

Using our previously published RNA-Seq datasets that examine the changes in gene expression that occur as oocytes enter cellular quiescence ¹⁶, we observed a significant 1.4-2.5 fold increase in the mRNA expression of 26 genes involved with the proteasome and the UPS systems. These genes include subunits for both the 20S core particle and the 19S regulatory particle (Extended Data Table 1), suggesting that the observed increase in proteasome activity is driven by increased expression of the 26S holoenzyme.

“Provide controls for intact proteasome complexes. The proteasome activity assay buffer does not contain ATP. Accordingly, the 26S will most probably fall apart.”

-Thank you pointing out this error in our text. All buffers used for mitochondrial isolation, mitochondrial associated proteasome activity, zymography, K48 Ubiquitination, and native westerns contained 100 uM ATP. We apologize for the confusion. We also now include source data files showing full gel images where for the native gels and zymography assays showing the complexes are indeed assembled.

“Treatment of oocytes with 50 μM MG132 for 2 hours will alter mitochondrial function and is not a low dose proteasome inhibitor treatment as also shown by the authors in Ext Figure 1A. Alterations in mitochondrial function upon proteasome inhibition is a well-known phenomenon (e.g. first description: Sullivan et al., J. Biol. Chem 2004 May 14;279(20):20699-707) and does not prove that recruitment of proteasomes to the mitochondria drives mitochondrial remodeling. “

- We appreciate the reviewer’s insight and we have corrected the aforementioned text.
- We agree that reduced UPS function has been associated with reduced mitochondrial respiration. However, the cell type specific relationships between the UPS and mitochondria in *In vivo* models such as *Drosophila* remain an open question. Our work shows that the programmed loss of mitochondrial membrane potential we reported (Sieber et al Cell 2016) during oogenesis is impaired by proteasome inhibition. These observations highlight a new link between the UPS and developmental remodeling of mitochondrial function.

“ The figure legends lack any information on the number of independent experiments, statistics details on the experiments and need to be significantly improved. Please clearly show where technical or biological replicates were used”

We thank the reviewer for pointing this out. As mentioned earlier we have added text in figure legends throughout the manuscript to address this issue. We now also provide source data for

every figure and extended data figure. All data points presented in the manuscript reflect biological replicates.

“The study lacks mechanistical data on how the proteasome is recruited to the mitochondria. It appears to involve VDAC, but it is not clear how that might work.”

- We agree this is an important point and now provide evidence that VDAC is polyubiquitinated in quiescent oocytes consistent with a link to the UPS and protein turnover. We also now provide experiments where we purify proteasomes using a RPT6-Flag transgene from quiescent cells and we observe an enriched interaction with mouse VDAC1. While we do see some background binding in our control IP the interaction is much stronger in the RPT6-flag purification of proteasome. Consistent with these results GSK3 phosphorylation of VDAC displays a much milder effect on VDAC turnover than what we observe with TOM22. Taken together these data suggest VDAC phosphorylation induces proteasome recruitment and once recruited to the mitochondrial surface a subset of VDAC, perhaps unbound by the proteasome, is turned over by the UPS. These data suggest that VDAC plays a crucial role in the recruitment of the proteasome to the mitochondria.

“Conclusions are not always supported by the data. e.g. Figure 2: Reduced OCR via inhibition of FAO, activates proteasome activity associated with mitochondria and K48 ubiquitination. Vice versa, activation of OCR by inhibition of GSK inhibited proteasome activity. These data show the OCR activity and proteasome activity are coupled and not “These data indicate that GSK induces mitochondria-associated proteasome activity during quiescence, in part, through inhibition of the FAO pathway.” (line 131ff). Similar in line 212ff: “VDAC phosphorylation by GSK3 triggers the recruitment of the proteasome to the mitochondria and promotes the turnover of mitochondrial outer membrane proteins.” This conclusion is not supported by the data which merely show that VDAC is required for increasing proteasome activity associated with mitochondria when this is activated by silencing FAO genes. The data do not show that phosphorylation by GSK is crucial in that process. GSK3 is able to phosphorylate VDAC but whether this is required for its effect on proteasome activity is not shown. Again in line 253ff: “activity is significantly lower and proteasome recruitment to the mitochondria further compromises cytosolic protein turnover rate.” The authors merely show distinct proteasome activities in total versus mitochondrial fraction. One cannot deduce causality from that observation.”

- The reviewer is correct and we have reworked the discussion of the manuscript to reflect more conservative assessment of the data.
- Regarding 131ff --we now state that “These data suggest that during quiescence GSK3 regulates the stability of mitochondria fatty acid oxidation proteins and, in turn, suppression of fatty acid oxidation promotes proteasome recruitment to the mitochondria”

- Regarding 212ff --we have also added additional data regarding VDAC phosphorylation and its potential interaction with the proteasome and now state that "Taken together these data suggest during quiescence GSK3 triggers proteasome recruitment and VDAC functions recruit the proteasome to the surface via a potential direct interaction. Once recruited to the mitochondrial surface a subset of VDAC is turned over by the UPS."
- Regarding 253ff --we have added additional data that shows induced forms of quiescence display a 30-50% decrease in cytosolic proteasome activity using *Drosophila* and NIH 3T3 cells. We now state " Consistent with this model we measured cytosolic proteasome activity in developing egg chambers from fed and amino-acid starved females and found a roughly 60% reduction in cytosolic proteasome activity consistent with the increased aggregation of mei31B further supporting the idea that induced forms of quiescence display compromised proteostasis. Moreover, this work is supported by recent studies that have suggested a direct link between mitochondrial metabolism and UPS regulation^{20,21}."

"The authors do not investigate what functional consequence the 26S recruitment to mitochondria has. This is a weakness of the study. "

- We have provided a substantial amount of data assessing of how all of the factors (GSK3, VDAC, MTPa and ETFa) in this study impact mitochondrial metabolism in quiescent oocyte.

In our previous publication Sieber et al 2016 we show that GSK3, a key regulator of proteasome recruitment to the mitochondria, promote mitochondrial proteins turnover, induces the suppression of ETC activity, maintains mitochondria membrane potential, promote nutrient storage in mature oocytes. In this manuscript, we provide that data examines the impact of VDAC on: proteasome recruitment to the mitochondria, mitochondrial respiration, mitochondrial membrane potential, and the metabolite profile of quiescent eggs. We also examine the impact of FAO pathway on mitochondrial membrane potential and respiration. Moreover, the phenotypes we observe in all of these studies consistently support our proposed model.

However, to attempt examine this in greater detail we attempted to rescue the VDAC mutant using a phospho-resistant VDAC transgene and then examine how this impact mitochondrial function. However, the phospho-resistant form of VDAC failed to rescue the female reproductive defects in VDAC mutant animals. Due to the lack of mature stage 14 oocytes we were unable to examine this any further. We now also provide source data for every figure and extended data figure. We also attempted to examine the functions of VDAC, TOM complex, and GSK3 in our *Neurospora* model and found that deletion of these gene caused defects in growth, sporulation, and germination making it impossible to us this system to examine this in further detail.

Figure 3: What happens to proteasome activity and recruitment to mitochondria in GSK silenced/VDAC het cells in a similar experiment as Figure 3J.

- We agree this would be quite interesting. We attempted this experiment and observed germline developmental defects that precluded further examination of this genetic interaction.

“12. Discussion is far-fetched and does not include any published data on the interplay of mitochondria and proteasome. It is well established that the proteasome is localized to mitochondria. The interplay between mitochondrial respiratory chain function and proteasome activity is also well established but not mentioned by the authors”

- We agree with the author and now provide additional text and citations throughout the manuscript that discusses the role of the proteasome in mitochondrial quality control and the impact of mitochondrial on proteasome activity.
- While we agree that components of the UPS, such as F-box proteins and Ubiquitin ligases, are known to localize to the mitochondria we have found no publications that show the intact functional proteasome is recruited to the mitochondrial surface or measure proteasome activity in mitochondrial fractions. We have however added additional text to mention that components of the UPS do localize to the mitochondria.

Reviewer #2

We appreciate reviewer# 2 comments they have aided improving the clarity and transparency of the manuscript. We have adjusted our conclusions and provide more experimental detail in the methods section and figure to address their concerns.

Original comments

Reviewer #2 (Remarks to the Author):

Summary

*The proper regulation of quiescence is critical for many biological processes, including tissue homeostasis and oocyte development. Previous work by the senior author established *Drosophila* oogenesis as a system to study mitochondrial respiratory quiescence (MRQ), showing that suppression of insulin signaling/activation of GSK3 triggers MRQ and glycogen accumulation in the mature oocyte. In this study, Yue et al. investigate the molecular mechanisms linking GSK3 activation and MRQ in *Drosophila* oocytes, and their evolutionary conservation in mammalian cells and fungi using a combination of genetics, cell biology, cell fractionation, biochemistry, proteomics and metabolomics approaches. The authors show that the expression and activity of the ubiquitin-proteasome system (UPS) and K48 ubiquitination (followed by turnover of ubiquitinated proteins) increases as oocytes enter quiescence, and the levels of proteins involved in fatty acid oxidation are reduced. Conversely, knockdown of GSK3 in oocytes results in elevated mitochondrial activity and increased levels of proteins involved in fatty acid oxidation. Consistent with their model that GSK3 inhibits fatty acid oxidation and trigger proteasome recruitment to mitochondria, knockdown of mitochondrial fatty acid oxidation genes led to increased levels of mitochondria-associated proteasome activity and increased K48 ubiquitination. Using proximity labeling combined with cell fractionation, the authors identified GSK3-associated proteins and showed that VDAC (a mitochondria porin ion channel in the outer mitochondrial membrane), and components of the TOM*

complex (a protein complex in the outer mitochondrial membrane involved in protein translocation into mitochondria) are directly phosphorylated by GSK3 *in vitro*. They also showed that overexpression of GSK3 (but not a kinase dead mutant) in 293T mammalian cells leads to very reduced levels of Tom22 and VDAC, and they also identified the specific phosphorylation site in VDAC required for this effect. They propose the model that GSK3 recruits the proteasome to the mitochondria by directly phosphorylating Tom22 and VDAC at the outer mitochondrial membrane and by repressing fatty acid oxidation in a VDAC-dependent manner. Remarkably, the authors show similarities between developmental quiescence in *Drosophila* oocytes and other types of induced or developmental quiescence. They show that amino acid deprivation induces precocious MRQ in developing *Drosophila* follicles (based on TMRE staining). They also show that serum deprivation-induced quiescence in mouse fibroblasts leads to reduced mitochondrial activity (and of mitochondrial proteins), increased glycogen levels, and proteasome recruitment to the mitochondria, and that developmentally quiescent *Neurospora crassa* spores also have elevated total and mitochondria-associated proteasome activity, demonstrating the evolutionary conservation of MRQ mechanisms.

Critique

The findings reported in this study represent a significant advance in our understanding of highly conserved mechanisms underlying quiescence, and should be of interest to a wide range of scientists. Notably, the authors combine a variety of powerful experimental approaches and distinct experimental systems to address important questions, test novel hypotheses, and open new directions of thought in the field. However, there are some concerns that should be addressed by the authors regarding methodological details, clarification for some of their data/conclusions, and the need to soften a couple of their conclusions, as outlined below.

Specific points:

1) Throughout the methods, more information is needed to demonstrate rigor and reproducibility. For every single biochemical, genetic, microscopy, etc, experiment, the authors should clearly specify how many times the experiments were repeated, sample sizes, and statistical methods used to analyze the data. More details are also needed for some of the experiments to ensure sufficient information for reproducibility. For example, in line 587, it is not clear what "sufficient" means precisely. Under "TMRE staining", clarify if experiment with MG132 has insulin added to PBS as well (lines 599-600). For all construct cloning, include name of cDNAs used and where they were obtained, provide all primer sequences in a table, explain how mutations were introduced, etc. For mitochondrial isolation/native PAGE assay, explain number of staged oocytes used (line 625) and how oocytes were homogenized – motorized pestle? FastPrep machine? (line 627). For western blotting analysis,

include what buffer the samples were in prior to loading onto denaturing versus native gels, where gradient gels were purchased, composition of transfer buffers and of TBST, specify secondary antibodies and source (also for primary antibodies), etc. Similarly, include more details for Seahorse assays, *in vitro* kinase assays (including vectors for targets), generation of APEX constructs and fly lines, etc.

2) Extended data Fig. 2C legend: authors should explain more precisely what criteria were used for the classifications of normal, moderate and severe.

3) Extended data Fig. 3A: the authors did not include the expression data for the APEX fused to CD8 control.

4) There is a general concern that all RNAi experiments were done using single RNAi lines.

5) In Fig. 3E (and lines 176-177), the overexpression of the kinase dead GSK3 is sufficient to reduce the levels

of Tom22 and VDAC (albeit to a much lesser extent than the wild type GSK3) in 293T cells. The authors should briefly discuss possible reasons for that. (e.g. Is it possible that GSK3 has some kinase-independent roles?)

6) The authors should be extra clear in their explanation of a seemingly confusing but interesting aspect of their model/data: VDAC phosphorylation by GSK3 appears to trigger proteasome recruitment to the mitochondria and subsequent degradation of VDAC in 293T cells (see lines 168-183). Yet, knockdown of VDAC has the same phenotype as GSK3 knockdown (which should lead to MORE stable VDAC) in *Drosophila* oocytes (Fig. 3H, I), and VDAC loss of function dominantly suppresses the increased proteasome activity induced by knockdown of fatty acid oxidation genes (Fig. 3J). These data seem to imply that VDAC plays an important role in inducing MRQ prior to its degradation by the proteasome. [Although VDAC knockdown paradoxically reduces oxygen consumption and increases glycogen levels at the same time that it increases TMRE levels (Extended data Fig. 6A-C) and reduces mitochondria-associated proteasome activity (Fig. 3I)!?] A clarification of these confusing points would be very helpful to the readers.

7) Lines 227-233: in these sentences, the author should soften their conclusions regarding the mechanisms involved in amino acid deprivation-induced quiescence because they did not directly investigate proteasome recruitment to mitochondria in this system. The tone of their conclusions should be adjusted accordingly.

8) The conclusion in lines 261-263 is similarly overstated.

9) Regarding some key aspects of their model, the authors should consider testing whether overexpression of key fatty acid oxidation enzymes in the *Drosophila* germline prevents mature oocyte MRQ (and recruitment of the proteasome to mitochondria). They should also speculate/discuss how they envision the proteasome recruitment to the mitochondria in their model: does it associate with the outside of mitochondria, or does it get imported somehow into the mitochondria? They should also draw a connection of their study with published work on the import of proteins into mitochondria and whether mitochondria import might be necessary for proteasome recruitment to mitochondria.

10) The authors should consider subdividing their manuscript into different sections with subtitles to improve the flow and clarity of the manuscript.

Typos, etc:

- Line 18: M.H.S. instead of M.S.?
- Lines 33-34: sentence should be re-written to make more sense grammatically and otherwise.
- Line 136: incorrect figure citation
- Line 187: authors should clarify more precisely what they mean by this sentence and provide a reference citation.
- Line 198: reference citation missing for FAO-related metabolites
- Line 204: data "show"
- Lines 218-219: clarify in which organisms/systems this has been previously shown.
- Line 318: typos
- All figure legends are missing a figure title, and the figure legends are somewhat cursorily written.
- In figure 2, the order of panels does not correspond to the figure legend.
- In figure 4G legend, "experimental strategy" does not seem appropriate because they are just showing the "lifecycle" of *N. crassa*.

- Extended data Fig. 4D: why not include Mut1 as a comparison as well?
- Line 582: small case for crassa
- Line 650: 1B1 (not anti-1B1)
- Line 664: instead of "gels" were transferred, "proteins" were transferred.
- Line 666: A ":" is missing after "follows"
- Lines 682-683: instead of "the rest experimental processes", "the remaining experimental procedures"
- Edit for additional typos.

Response to specific comments"

"Throughout the methods, more information is needed to demonstrate rigor and reproducibility. For every single biochemical, genetic, microscopy, etc, experiment, the authors should clearly specify how many times the experiments were repeated, sample sizes, and statistical methods used to analyze the data. More details are also needed for some of the experiments to ensure sufficient information for reproducibility. For example, in line 587, it is not clear what "sufficient" means precisely. Under "TMRE staining", clarify if experiment with MG132 has insulin added to PBS as well (lines 599-600). For all construct cloning, include name of cDNAs used and where they were obtained, provide all primer sequences in a table, explain how mutations were introduced, etc. For mitochondrial isolation/native PAGE assay, explain number of staged oocytes used (line 625) and how oocytes were homogenized – motorized pestle? FastPrep machine? (line 627). For western blotting analysis, include what buffer the samples were in prior to loading onto denaturing versus native gels, where gradient gels were purchased, composition of transfer buffers and of TBST, specify secondary antibodies and source (also for primary antibodies), etc. Similarly, include more details for Seahorse assays, in vitro kinase assays (including vectors for targets), generation of APEX constructs and fly lines, etc."

- We have provide more detail to provide clarity into our experimental methods. Moreover, we have also added additional detail regarding experiment number and statistical methods into the figure legends to make data interpretation more transparent. We have modified our tables and graphs to show individual data points. We now also provide source data for every figure and extended data figure.

3) Extended data Fig. 3A: the authors did not include the expression data for the APEX fused to CD8 control.

- We apologized for the confusion. CD8-apex was originally characterized in a previous publication. However, the sentence referencing that publication was accidentally deleted when shortening the original manuscript. We have added the citation back into the main text.

4) "There is a general concern that all RNAi experiments were done using single RNAi lines."

- We understand the reviewer's concern. Due to challenges in transgene expression in germ cells many RNAi transgenes do not express well in the germline. Prior to studying these genes we screened multiple RNAi transgenes for each gene and found that the transgenes used in this manuscript provided the most effective silencing of target gene expression without causing severe developmental defects in the germline. We now provide Q-PCR data showing the efficiency of each knockdown. This is also why we targeted multiple steps in the mitochondrial fatty acid oxidation pathway and used the heterozygous VDAC^{rev8} allele in our epistasis experiment with the fatty acid oxidation genes. We also attempted to validate these results with mutations in GSK3, VDAC, and ETFa. Mutations in GSK3 and ETFa are lethal, as reported in the literature, and mutations in VDAC are semi-lethal and display defects in oogenesis that make collecting quiescent egg samples for biochemical analysis impossible.

In Fig. 3E (and lines 176-177), the overexpression of the kinase dead GSK3 is sufficient to reduce the levels of Tom22 and VDAC (albeit to a much lesser extent than the wild type GSK3) in 293T cells. The authors should briefly discuss possible reasons for that. (e.g. Is it possible that GSK3 has some kinase-independent roles?)

- Based on our proximity labeling experiments we found that GSK3 associates with UPS components as significantly as it does with mitochondrial outer membrane proteins. However, these UPS proteins are not phosphorylated by GSK3. This may suggest that GSK3 functions in larger complex to aid in proteasome recruitment beyond its kinase activity. We are also studying other signaling pathways that control MRQ and it's possible that expressing the Kinase dead version of GSK3 may cause compensation from these other pathways that lead to partial turnover of Tom22 and VDAC. Given the multiple interpretations of this data we withheld further discussion to prevent confusion. If necessary we are happy to add in additional text.

"The authors should be extra clear in their explanation of a seemingly confusing but interesting aspect of their model/data: VDAC phosphorylation by GSK3 appears to trigger proteasome recruitment to the mitochondria and subsequent degradation of VDAC in 293T cells (see lines 168-183). Yet, knockdown of VDAC has the same phenotype as GSK3 knockdown (which should lead to MORE stable VDAC) in Drosophila oocytes (Fig. 3H, I), and VDAC loss of function dominantly suppresses the increased proteasome activity induced by knockdown of fatty acid oxidation genes (Fig. 3J). These data seem to imply that VDAC plays an important role in inducing MRQ prior to its degradation by the proteasome. [Although VDAC knockdown paradoxically reduces oxygen consumption and increases glycogen levels at the same time that it increases TMRE levels (Extended data Fig. 6A-C) and reduces mitochondria-associated proteasome activity (Fig. 3I)!?] A clarification of these confusing points would be very helpful to the readers."

- The reviewer is correct in their assessment regarding the role of VDAC in proteasome recruitment. We now provide data that suggests VDAC directly associates with the proteasome to facilitate recruitment (extended data figure 5) and once recruited we believe a subset of VDAC, possibly unbound by the proteasome, is turned over. We have added additional text to discuss this model in greater detail.

“7) Lines 227-233: in these sentences, the author should soften their conclusions regarding the mechanisms involved in amino acid deprivation-induced quiescence because they did not directly investigate proteasome recruitment to mitochondria in this system. The tone of their conclusions should be adjusted accordingly. 8) The conclusion in lines 261-263 is similarly overstated.”

- We agree with the reviewer and have provided a more conservative interpretation of the data.

“9) Regarding some key aspects of their model, the authors should consider testing whether overexpression of key fatty acid oxidation enzymes in the Drosophila germline prevents mature oocyte MRQ (and recruitment of the proteasome to mitochondria). They should also speculate/discuss how they envision the proteasome recruitment to the mitochondria in their model: does it associate with the outside of mitochondria, or does it get imported somehow into the mitochondria? They should also draw a connection of their study with published work on the import of proteins into mitochondria and whether mitochondria import might be necessary for proteasome recruitment to mitochondria.”

- We agree that such overexpression experiments would be interesting however beginning in stage 10 global transcription and translation has shut down so expressing transgenes in stage 14 oocytes are not possible. Moreover looking at the transition periods (stage 11-13) are very challenge given those stages progress every quickly. So a given female may only contain few stage 11-13 egg chambers making biochemical analysis of these stages impossible.

-Based on our ongoing studies we believe that the proteasome is recruited to the outer mitochondrial membrane and that through retro-translocation of inner mitochondrial proteins are turnover over. It is also possible that the elimination of mitochondrial outer membrane protein may activate inner mitochondrial proteases to remodel the inner membrane. We have added additional text to the manuscript to discuss this.

Reviewer #3

Original comments

Review of: Highly conserved shifts in ubiquitin-proteasome system (UPS) 2 activity drive mitochondrial remodeling during quiescence 3 4 Authors: Sibiao Yue¹, George N. DeMartino¹, FangZhou Zhao¹, Yi Liu¹ 5 , Matthew H. Sieber^{1*}

This is an excellent paper concerning the cell quiescence, a state found in germ cells and other cells that are deeply prevented from proliferation and differentiation. The authors chose an excellent system the fly ovariole, which houses oocytes that will stored in a stable state until hormonally stimulated to mature and be ovulated. The transition to that state is an important problem and likely to be general and the Drosophila ovary is a great system. In the quiescent state the mitochondria are also quiescent in a state known as mitochondrial respiratory quiescence MRQ and these authors see this as primary. They provide excellent proteomic(MS) and other evidence that proteasome activity associated with the mitochondria is more active. The implicate GSK3 activity in phosphorylating outer membrane proteins and particularly VDAC a anion transporter that brings in much of the metabolic intermediates. They show that this is a key

step driving quiescence and hormonally sensitive. What is truly fascinating here is that energy metabolism is regulated by proteolysis and this in turn is regulated by proteasome localization. If there is one quibble is that the dynamics should be visualizable at the light microscope level. They argue that there is depletion of the proteasome from the cytosol. Microscopic monitoring would allow the kinetics to be better determined, to note if localization is spatial on a single mitochondrion or whether it only affects particular mitochondria. It would be possible to do a better dose response. This could be a static experiment with anti proteasome antibodies or perhaps a dynamic experiment with fluorescent markers. I do not think this is essential as there is convincing enough data on mitochondrial state.

I find little to complain about here and at a high level I am sure that this will be important. It is an unusual problem. It is an important problem. It is mechanistically cleanly presented. The tests of necessity and sufficiency are clear. Of course there may be several ways of initiating MRQ and several different outcomes. The authors followed the VDAC channel but it is not clear if it is the only path even though it looks necessary.

Response to specific comments

“Microscopic monitoring would allow the kinetics to be better determined, to note if localization is spatial on a single mitochondrion or whether it only affects particular mitochondria. It would be possible to do a better dose response. This could be a static experiment with anti proteasome antibodies or perhaps a dynamic experiment with fluorescent markers. I do not think this is essential as there is convincing enough data on mitochondrial state.”

- We appreciate this reviewer's comments and enthusiasm for this manuscript. The reviewer's suggestion regarding visualizing this phenomena by microscopy is interesting. We have attempted this with several proteasome subunit antibodies (such as PSMB1, PSMC4, and RPT6), consistent with the literature, core subunits antibodies tested display expression exclusively in the nucleus while many cap factors such as RPT6 express everywhere except the nucleus. In contrast, PSMC4 expresses everywhere in the cell except the nucleolus. (for examples see the Human Protein Atlas). The lack of uniformity in the localization of these proteasome proteins makes visualization of this phenomena difficult. However, to address this weakness we purified cytosolic fractions and now show that a 30-50% reduction in cytosolic proteasome activity in induced forms of quiescence (in both drosophila and NIH3T3 cells.)

Reviewer #4

Original comments

“This manuscript describes the authors' studies investigating the role of the ubiquitin-proteasome system in driving mitochondrial remodeling during quiescence. The authors argue that an elevated number of proteasomes in quiescent cells and their increased recruitment to mitochondria results in a phenomenon termed mitochondrial respiratory quiescence. More specifically, GSK3 was found to trigger proteasome recruitment to mitochondria. GSK3, through outer membrane proteins including VDAC, affects proteasome

degradation. These findings were reported in *Drosophila*, mouse and fungi.

Comments

The authors may have elucidated a pathway involving fatty acid oxidation affecting proteasome activity through VDAC. There are some data that suggest that this pathway may exist. Fig 3J does suggest that VDAC may be involved in a pathway in which fatty acid oxidation affects proteasome activity. However, there are many concerns about the data and conclusions at this point. The authors are trying to establish a very complex model. At every point, concerns about rigor make it difficult to agree with the authors conclusions. Many important controls and information needed to agree with the authors conclusions are missing. The authors are trying to make an argument about a pathway that is active in quiescence, but most of the figures do not show proliferating and quiescent cells so the selectivity for quiescence is not clear. More rescue experiments would be needed to demonstrate causality in the signaling pathway the authors hypothesize. For instance, there is no evidence that VDAC phosphorylation, rather than simply VDAC levels, is involved in this pathway. Further, the work is almost exclusively performed in *Drosophila*, but the authors seek to make very broad claims about this pathway being consistent in all quiescent cells that are not justified.

The reduced mitochondrial oxidation of fatty acids that the authors argue is a hallmark of quiescence is not consistent with the literature. For instance, in Ito et al (Nat Med 2012), the authors found that inhibiting fatty acid oxidation in hematopoietic stem cells resulted in exit from quiescence. The authors should review this literature further and frame their manuscript accordingly.

The authors need to explain the *Drosophila* oocyte model: what sample are being taken and when and what is considered quiescent and why?

Fig 1b: The figure legend should explain what is being shown here. How many independent samples from how many flies were used to generate these data? How many times was this performed? What does St1-8, St14, 0-2h and 16-20h refer to?

In general, this information should be added to all of the figure legends.

Line 72: by what criteria were these genes "significant" in the RNA seq data? What samples are being compared to define significance?

Line 77: what does "onset of quiescence" mean? How was this modeled in *Drosophila*?

What are we comparing in 1D? Is there quantification on multiple samples that would convince us that these two lanes are different?

Fig 1E: what controls were performed to confirm that mitochondria were isolated for these experiments?

Where are the data to support this statement: "Moreover, we observed that K48 ubiquitination returns to normal in the hours after the onset of quiescence, suggesting that these ubiquitinated proteins have been turned over."?

Are these proteasomes inside the mitochondria or attached to them on the outside?

1G: Are these mitochondrial proteins being ubiquitinated? Can the authors show any examples?

Fig 2B: GSK3 inhibition affects oocytes as expected from the literature, but these results don't make it clear that there is a quiescence-specific effect

Fig 2B and 2C: Where is the validation of GSK3 RNAi knockdown? It would be best to have more than 1 siRNA to ensure the results aren't off target effects. This is true for all of the RNAi experiments.

Fig 2H and 2I: are these performed in quiescent cells? How do quiescent and proliferating cells compare for these assays?

Extended data Fig 3: It's clear that mitochondrial proteins were pulled down, but it is not clear how many non-mitochondrial proteins were pulled down. What fraction of the proteins are mitochondrial?

Fig 3G: what are we looking at in this figure? What is the difference between lanes 1 and 2? Lanes 3 and 4? Are they two examples of the same thing? Have these data been quantified? Is there a statistically significant difference?

Fig 3J: why use heterozygous VDAC lines?

These knockout and knockdown models do not demonstrate that VDAC phosphorylation, as shown in the schematic, is important for proteasome activity.

Fig 4 If the authors would like to make the argument that cytosolic proteasome-mediated protein degradation is being compromised during quiescence, it is suggested that they use a better model for monitoring cytoplasmic proteasome activity. It is also suggested that they use additional measures for the presence of protein aggregates in addition to Me31B. It is not possible for the reader to agree that there is a systemic effect on protein degradation from these data alone.

All figures: why is it that when cells have higher levels of proteasome activity, K48 ubiquitination is higher?

Response to specific comments

We appreciate the comments of reviewer#4 and have provide text changes and additional data to address their concerns

“ The authors are trying to establish a very complex model. At every point, concerns about rigor make it difficult to agree with the authors conclusions. Many important controls and information needed to agree with the authors conclusions are missing.

-We have added addition data regarding the proposed mechanism. We have also added additional controls and more indepth experimental details to address the concerns the reviewer.

“The authors are trying to make an argument about a pathway that is active in quiescence ,but most of the figures do not show proliferating and quiescent cells so the selectivity for quiescence is not clear”

- We are sorry if there is confusion in figures 1 and 5 we do show a specific increase in proteasome activity during quiescence in Fungi, Drosophila, and mammalian cells. With regards to figures 2 and 4 where we examine the impact to of GSK3, VDAC, and FAO on proteasome recruitment in quiescence oocytes we are by limited material in these studies. Each experiment presented requires the dissection of hundreds of ovaries and the manual collection of staged oocytes. So, as a result, to maintain high sample quality we examined impact of GSK3 and VDAC specifically in quiescent cells. Active egg chambers are considerably smaller and have 3-4X fewer mitochondria than quiescent eggs so we limited in our capacity use these samples. Moreover mitochondrial associated proteasome activity is very low in growing oocytes so our ability to examine GSK3 and VDAC phenotypes in active oocytes is limited.

“More rescue experiments would be needed to demonstrate causality in the signaling pathway the authors hypothesize.”

-we attempted to rescue the VDAC mutant using a phospho-resistant VDAC transgene and then examine how this impact mitochondrial function. However, the phosphor-resistant VDAC rescue displayed defects germline line development preventing further study.

“For instance, there is no evidence that VDAC phosphorylation, rather than simply VDAC levels, is involved in this pathway.”

- Based our data reduced VDAC levels do not promote proteasome recruitment to the mitochondria. Reducing VDAC levels using RNAi and heterozygous mutations inhibits proteasome recruitment figure 4. Moreover, we have provided new data that shows VDAC has an enriched association with the proteasome in quiescent cells. We have also reworded the text to reflect this new interpretation.

“ Further, the work is almost exclusively performed in Drosophila, but the authors seek to make very broad claims about this pathway being consistent in all quiescent cells that are not justified.”

Due to limitations of the systems many experiments in this manuscript were not possible in mammals or yeast. For example, we attempted functional studies in neurospora and found VDAC deletions display growth defects and defective sporulation. Whereas mutations in the GSK3 homolog (OR74A) display defective sporulation consistent with the known roles for GSK3 in meiosis and fungal sporulation. In mammalian cells, there has been duplication of VDAC and the genome contains 3 direct VDAC orthologs (VDAC1-3) making functional studies challenging. Moreover, treatment of cells with GSK3 specific inhibitors BIO-acetoxime and 1-Azakenpaullone killed the quiescent cells preventing their analysis. We have provided substantial evidence that the proteasome recruitment to the mitochondria is conserved. We have adjusted our text to focus our discussion of conservation to the proteasome recruitment to the mitochondria.

“The reduced mitochondrial oxidation of fatty acids that the authors argue is a hallmark of quiescence is not consistent with the literature. For instance, in Ito et al (Nat Med 2012), the authors found that inhibiting fatty acid oxidation in hematopoietic stem cells resulted in exit from quiescence. The authors should review this literature further and frame their manuscript accordingly”

- The reviewer is correct and we never meant to imply that every aspect of quiescence we discuss in this manuscript is conserved in every system. While we never specifically say that the role of FAO in proteasome recruitment is conserved we agree that with the reviewer that as worded it could be confusing. The effect of FAO in this process is likely dependent on the local environment and tissue lineage. We have added additional text to clarify that the role of FAO in proteasome recruitment to the mitochondria in other populations of quiescent cells remains an open question.

“The authors need to explain the Drosophila oocyte model: what sample are being taken and when and what is considered quiescent and why?”

- During stage 1-10 of oogenesis egg chambers are growing and display high levels of transcription and translation. Beginning in stage 10 transcription and translation decline dramatically and the oocyte become completely quiescent in stage 14. We have added text to make this clear in the beginning of the results section.

“Fig 1b: The figure legend should explain what is being shown here. How many independent samples from how many flies were used to generate these data? How many times was this performed? What does St1-8, St14, 0-2h and 16-20h refer to?”

- We appreciate the reviewer pointing this out and we have added additional text to the figure legends throughout the manuscript to provide much more experimental information and a clear description of the number of samples assessed in each figure.

Line 72: by what criteria were these genes “significant” in the RNA seq data? What samples are being compared to define significance?

- We apologize for omitting this data and we have added the requested information to the figure legend. These samples compare early egg chambers to stage 10 oocytes which are entering quiescence. Stage 10 was chosen due to the fact that at stage 14 there is no transcription or translation in these cells. All genes highlighted in that data display a FDR adjusted pvalue of less than .05. This text has been added to the figure legend.

Line 77: what does “onset of quiescence” mean? How was this modeled in Drosophila?

- We apologize for the confusion. The onset of quiescence simply refers to timepoint in oogenesis when the oocyte stops global transcription and translation and become metabolically inactive. We have tried to make that point more clearly in the text.

“Fig 1E: what controls were performed to confirm that mitochondria were isolated for these experiments?”

- An example of the purity of our mitochondria isolations can be seen in extended data figure 2D.

“Where are the data to support this statement: “Moreover, we observed that K48 ubiquitination returns to normal in the hours after the onset of quiescence, suggesting that these ubiquitinated proteins have been turned over.”

- As stated in the text these data are shown in (extended data figure 1 D). In this experiment we use nutrient deprivation and isolation from males to force eggs to be stored in a quiescent state for 24hr and 48hrs. In this experiment we observed a decline in K48 ubiquitination in mitochondrial fractions as oocytes are held in a quiescent state. We have added information to the figure legends to make this point more clearly.

“Are these proteasomes inside the mitochondria or attached to them on the outside?”

- Transport into the mitochondria is highly regulated and there are no known mechanisms for the transport for large complexes such as the proteasome into the mitochondria. We have no evidence that supports the idea that the proteasome can enter the mitochondria. As a result we find it highly unlikely that the proteasome enters the mitochondria but instead is simply bound to the outer membrane. This likely supports mitochondrial remodeling by blocking mitochondrial protein import (Tom complex turnover) or participating of the turnover of proteins exported by retrograde transport.

“1G: Are these mitochondrial proteins being ubiquitinated? Can the authors show any examples?”

- We now show that VDAC is K48 ubiquitinated in Figure 4E. We also systematic increases in K48 ubiquitination in mitochondrial fraction in figure 1G, and extended data 1D)

“Fig 2B: GSK3 inhibition affects oocytes as expected from the literature, but these results don’t make it clear that there is a quiescence-specific effect”

- Based on our previous work we have only observed these types of mitochondrial phenotypes in oocytes. Somatic cell respiration does not decline in active somatic cells in response to insulin suppression. This also consistent studies in mammalian quiescent B cells where GSK3 promotes quiescence and inhibits mitochondria function and glycolysis. However, we make no claims that GSK3 cannot impact mitochondrial function in other contexts. GSK3 clearly has additional roles outside of quiescence.

“Fig 2B and 2C: Where is the validation of GSK3 RNAi knockdown? It would be best to have more than 1 siRNA to ensure the results aren’t off target effects. This is true for all of the RNAi experiments.”

- We understand the reviewer’s concern. Due to challenges in transgene expression in germ cells many RNAi transgenes due not express well in the germline. Prior to studying these genes we screened multiple RNAi transgenes for each gene and found that the

transgenes used in this manuscript provided the most effective silencing of target gene expression without causing severe developmental defects in the germline. We now provide Q-PCR data showing the efficiency of each knockdown. This is also why we targeted multiple steps in the mitochondrial fatty acid oxidation pathway and used the heterozygous VDAC^{rev8} allele in our epistasis experiment with the fatty acid oxidation genes. We also attempted to validate these results with mutations in GSK3, VDAC, and ETFa. Mutations in GSK3 and ETFa are lethal, as reported in the literature, and mutations in VDAC are semi-lethal and display defects in oogenesis that make collecting quiescent egg samples for biochemical analysis impossible.

“Fig 2H and 2I: are these performed in quiescent cells? How do quiescent and proliferating cells compare for these assays?”

- These experiments were done in growing egg chambers. We have added text to make this point clear. Due to the already high activity in quiescent oocytes in these experiments in quiescent cells did not yield substantial difference upon FAO inhibition

“Fig 3G: what are we looking at in this figure? What is the difference between lanes 1 and 2? Lanes 3 and 4? Are they two examples of the same thing? Have these data been quantified? Is there a statistically significant difference?”

- We apologize if this was not clear. Lane one is a control showing VDAC transgene expression, lane 2 shows that GSK3 expression reduced the levels of VDAC roughly 30%. Lane 3 shows the expression level of VDAC allele containing a mutated phosphorylation site. Lane 4 shows the mutated version of VDAC is no longer impacted by GSK3 expression. As stated in the text quantification of this data is located in extended fig 4D.

“Fig 3J: why use heterozygous VDAC lines?”

- We apologize for the confusion. This is a common genetic approach in *Drosophila* to investigate dominant genetic interactions. Homozygous mutations in VDAC have severe defects in egg development. By using the VDAC/+ hets we can bypass these developmental defects while at the same time assessing whether VDAC haplo-insufficiency can suppress the increased proteasome recruitment to the mitochondria we observe upon the suppression of FAO.

“These knockout and knockdown models do not demonstrate that VDAC phosphorylation, as shown in the schematic, is important for proteasome activity.”

- We have attempted to test the role for VDAC phosphorylation by rescuing the VDAC mutant with phospho-resistant VDAC transgenes. However, these lines did not completely rescue the VDAC mutant developmental phenotypes and displayed oogenesis defects that prevent the testing of this model. This cannot be simply tested in our cell culture model because there are multiple VDAC homologs in mice and they are known to be functionally redundant. In *Neurospora* we found that VDAC mutants display defects in growth, sporulation, and germination that prevented our ability to test this in that system.

“Fig 4 If the authors would like to make the argument that cytosolic proteasome-mediated protein degradation is being compromised during quiescence, it is suggested that they use a better model for monitoring cytoplasmic proteasome activity. It is also suggested that they use additional measures for the presence of protein aggregates in addition to Me31B. It is not possible for the reader to agree that there is a systemic effect on protein degradation from these data alone.”

- We now provide new data measuring cytosolic proteasome activity and now show a 30%(3T3 cells) to 50%(flies) reduction of cytosolic proteasome activity in starvation induced forms of quiescence.

“All figures: why is it that when cells have higher levels of proteasome activity, K48 ubiquitination is higher?”

- This is observed because we are collecting the samples as the cells enter quiescence. As we show in extended data 1 C&D if quiescent oocytes are forced to be retained by nutrient deprivation and isolation from males we find that mitochondrial K48 levels are dramatically reduced over a period of 48hrs while mitochondria associated proteasome activity remains fairly stable. Similarly, we collected spores as soon as they were formed. This is consistent with a build-up of K48 ubiquitinated proteins at the onset of quiescence and that these proteins get turned over as cells are maintained in quiescence.

REVIEWER COMMENTS

Reviewer #1 (Remarks to the Author):

Review of the revised manuscript “Highly conserved shifts in ubiquitin proteasome system (UPS) activity drive mitochondrial remodeling during quiescence” by Yue et al.

The authors addressed most of my concerns regarding the analysis of proteasome function and added additional data which strengthens the manuscripts. However, I still have serious concerns regarding the interpretation of their findings.

Yue et al. state already in the title of their manuscript that alterations of the UPS drive mitochondrial remodeling. I find this causality, however, questionable and not supported by the data. The authors very nicely show that during quiescence the activity of the 26S proteasome is increased within the mitochondrial fraction of oocytes and fibroblasts. They perform several interference studies using silencing of GSK3, VDAC, EFTA and MTPa and observe altered proteasome activity. However, all these interferences contribute to altered mitochondrial activity as either shown by Seahorse experiments or TMRM staining. Improved mitochondrial respiratory function is generally associated with lower proteasome activity while mitochondrial dysfunction is associated with elevated 26S activity. One plausible explanation would be that the proteasome is recruited to dysfunctional mitochondria via GSK3/VDAC as part of cellular mitochondrial quality control. The crucial role of the proteasome in mitochondrial quality control has been shown before. The authors do not take this explanation into consideration but suggest an opposite causality, i.e., that proteasome activity drives mitochondrial remodeling. The only experiment they show in support of that hypothesis comes from a proteasome inhibitor experiment. The authors observe mitochondrial alterations upon blocking proteasomal protein degradation. It is, however, well known, that blocking proteasome function induces cellular apoptosis and thereby affects mitochondrial function. Accordingly, it is not surprising that the high inhibitor doses used in this study effectively block protein degradation and cause severe mitochondrial dysfunction (50 micromolar MG132). This experiment does not provide any insight into the physiological role of the proteasome in quiescence and does not support the interpretation that proteasome dysfunction causes mitochondrial remodeling.

Other concerns:

1. please show effective silencing on the protein level not only RNA level.
2. Please show full native gel including 20S and 26S activity and blots thereof, not only the cut-out parts.
3. In quiescent oocytes proteasomal activity on mitochondrial fractions is elevated 12 fold compared to active oocytes. Assembly, however, is increased by about 2fold. How do you explain these discrepancies? Moreover, silencing experiments result in 0.25 -0.5 fold differences. Please explain how this would then be comparable to the condition in quiescent oocytes.

4. Please control that you have no ribosomes co-purifying with the mitochondria as these co-localize with the proteasome.

Reviewer #2 (Remarks to the Author):

Yue et al. have satisfactorily addressed most of my comments (other than a few remaining typos/grammar errors and lack of clarity in extended data Fig. 2C legend - Point 2 of my original review). I enthusiastically recommend publication of this revised manuscript in Nature Communications.

Reviewer #4 (Remarks to the Author):

The authors report an interesting mechanism involving a transition of the proteasome to mitochondria in quiescent cells with several different model systems. The revised manuscript is much stronger. The authors have responded to my questions and addressed my concerns.

Reviewer#1

"The authors addressed most of my concerns regarding the analysis of proteasome function and added additional data which strengthens the manuscripts. However, I still have serious concerns regarding the interpretation of their findings.

Yue et al. state already in the title of their manuscript that alterations of the UPS drive mitochondrial remodeling. I find this causality, however, questionable and not supported by the data. The authors very nicely show that during quiescence the activity of the 26S proteasome is increased within the mitochondrial fraction of oocytes and fibroblasts. They perform several interference studies using silencing of GSK3, VDAC, EFTA and MTPa and observe altered proteasome activity. However, all these interferences contribute to altered mitochondrial activity as either shown by Seahorse experiments or TMRM staining. Improved mitochondrial respiratory function is generally associated with lower proteasome activity while mitochondrial dysfunction is associated with elevated 26S activity. One plausible explanation would be that the proteasome is recruited to dysfunctional mitochondria via GSK3/VDAC to as part of cellular mitochondrial quality control. The crucial role of the proteasome in mitochondrial quality control has been shown before. The authors do not take this explanation into consideration but suggest an opposite causality, i.e., that proteasome activity drives mitochondrial remodeling. The only experiment they show in support of that hypothesis comes from a proteasome inhibitor experiment. The authors observe mitochondrial alterations upon blocking proteasomal protein degradation. It is, however, well known, that blocking proteasome function induces cellular apoptosis and thereby affects mitochondrial function. Accordingly, it is not surprising that the high inhibitor doses used in this study effectively block protein degradation and cause severe mitochondrial dysfunction (50 micromolar MG132). This experiment does not provide any insight into the physiological role of the proteasome in quiescence and does not support the interpretation that proteasome dysfunction causes mitochondrial remodeling.

Other concerns:

- 1. please show effective silencing on the protein level not only RNA level.*
- 2. Please show full native gel including 20S and 26S activity and blots thereof, not only the cut-out parts.*
- 3. In quiescent oocytes proteasomal activity on mitochondrial fractions is elevated 12 fold compared to active oocytes. Assembly, however, is increased by about 2fold. How do you explain these discrepancies? Moreover, silencing experiments result in 0.25 -0.5 fold differences. Please explain how this would then be comparable to the condition in quiescent oocytes.*
- 4. Please control that you have no ribosomes co-purifying with the mitochondria as these co-localize with the proteasome."*

We understand the reviewer's concern. However, we would like to point out that unlike other studies of mitochondrial quality control we are not causing non-specific mitochondrial dysfunction. Many previous studies use mutator cells, inhibitors, and disease-causing alleles to

drive mitochondrial dysfunction in studies of mitochondrial quality control mechanisms. However, we observe proteasome recruitment to the mitochondria normally during late oogenesis and neurospora sporulation. There is no mitochondrial dysfunction prior to proteasome recruitment in this context. We have examined this in flies in previous studies looking at ETC assembly, OCR, ETC activity, IFC, mitochondrial EM's, metabolomics, and mitochondrial proteomics (Sieber et al 2016, Hocaoglu et al 2021).

Additional comments in response to the reviewer#2

For example, mitochondrial dysfunction can be observed via metabolomics by a shift in glutathione oxidation state GSH/GSSG ratio. These metabolites are unaffected in our studies of the transition from active growth to quiescence in oocytes. However, in response to the reviewers continued concern I have attached a DHE (ROS specific stain) image of a stage 9 egg chamber. Stage 9 is the stage directly prior to the onset of quiescence and MRQ and while the posterior somatic follicle cells directly adjacent to the oocyte stain positively for ROS the oocytes is completely devoid of ROS signal suggesting that there is no obvious mitochondrial dysfunction prior to the period of proteasome recruitment. Our findings are consistent with the fact that oocytes eliminate dysfunctional mitochondria in a selective bottle-neck early in oogenesis to protect the oocyte from oxidative damage and ensure healthy embryogenesis <https://www.ncbi.nlm.nih.gov/pmc/articles/PMC6614061/>.

As a result there is no evidence to suggest that mitochondrial dysfunction precedes the onset of quiescence and mitochondrial recruitment of the proteasome.

Inhibiting GSK3 and VDAC cause reduced proteasome recruitment to the mitochondria during quiescence however GSK3-RNAi oocytes display substantially higher levels of respiration and VDAC-RNAi oocytes display significantly lower levels of respiration. This argues against the model for this being simply a mitochondrial quality control mechanism. Moreover, disruptions in the mitochondria quality control pathway cause complete sterility and manifest severe defects in early oogenesis unlike our manipulations of GSK3 and VDAC.

However, we now provide 2 additional pieces of data that strongly support our model.

First we show that overexpressing a phospho-resistant version of VDAC (VDAC-PR) is sufficient to block the proteasome recruitment to the mitochondria that occurs normally in quiescent oocytes (figure 4F). These data support the model where VDAC phosphorylation (not decreased respiration) is key factor in proteasome recruitment.

Second, we treated 3T3 cells with either 0.5uM rotenone or 0.5uM Antimycin A for 6 hrs and measured proteasome recruitment to the mitochondria. Consistent with our model we found that neither inhibitor had any effect on proteasome recruitment to the mitochondria (extended data figure 2F).

We have attempted other manipulation of the proteasome such as conditions expression of proteasome subunit dominant negative alleles and found the widespread germ cells death preventing further study. When examining the effect of MG132 we have to use high concentrations because the germ cells are surround by a dense muscle cell layer, a follicle cell epithelium, and a vitellin membrane that make permeability to the germ cell difficult. Moreover, the combination of MG132 and the mitochondrial inhibitors used in seahorse assays caused the cell to delaminate and prevented accurate OCR measurements.

Our current data shows that in controls there is a progressive loss of membrane potential as egg chambers progress through development and the MG132 treatment prevents this developmental loss of membrane potential. Moreover, it goes beyond the scope of this manuscript to examine all aspects of mitochondrial function that are impact by the proteasome in this context.

However, we have softened some of our statements regarding the impact of the proteasome on mitochondrial function.

1. Please show effective silencing on the protein level not only RNA level.

Most of the mitochondrial genes studied in this manuscript do not have available drosophila antibodies. We did obtain antibodies for the Drosophila porin/VDAC, however we did not see a band of the appropriate size. Moreover, given this is a transgenic RNAi approach which inhibits gene expression throughout oocyte development, and not transient inhibition RNAi as in cell

culture experiments, it is very typical of Drosophila studies to show Q-PCR data to quantify the effectiveness of RNAi experiments.

2. Please show full native gel including 20S and 26S activity and blots thereof, not only the cut-out parts.

We have provided full gel images of the zymography assays in the main text and native blots in source data.

2. In quiescent oocytes proteasomal activity on mitochondrial fractions is elevated 12 fold compared to active oocytes. Assembly, however, is increased by about 2fold. How do you explain these discrepancies? Moreover, silencing experiments result in 0.25 -0.5 fold differences. Please explain how this would then be comparable to the condition in quiescent oocytes?

-We are happy to explain this somewhat confusing point. As we show in extended data figure 1 mitochondrial associated proteasome activity accounts for 4% of total proteasome activity in in growing oocytes. During quiescence this increases ~12fold which would make it roughly 48% of the activity we observed in actively growing oocytes. However, during quiescence total proteasome activity increase 2-3 fold which would suggest the mitochondrial associated proteasome activity accounts for roughly 16-24% of the total activity seen in quiescent oocytes. In support of this idea starving females and inducing mitochondrial recruitment of the proteasome prematurely, prior to the developmental increase in proteasome number, shows a ~50% decrease in cytosolic proteasome activity. These data are consistent with recruiting the proteasome to the mitochondria in the absence of increase proteasome number.

Based on this idea if we examine the VDAC-RNAi experiments, mitochondria from VDAC-RNAi oocytes would recruit the proteasome at a significantly reduced level. instead of a 10-12 fold increase we would expect to see a 5-6 fold increase proteasome recruitment. In light of this, the 90% decrease in mitochondrial associated proteasome activity we observe in VDAC-PR expressing oocytes suggest while VDAC may be a key target for proteasome recruitment. It also suggest that another kinase or pathway may also work in parallel to GSK3 in this process.

4. Please control that you have no ribosomes co-purifying with the mitochondria as these co-localize with the proteasome."

It is unclear what bearing ribosomes would have on this manuscript and as a result go beyond the scope of this manuscript.. However, we have added additional controls to our fraction experiments. Given the major for contaminant for mitochondrial fractionation is membrane fractions such as plasma membrane and ER we examined their presence in our mitochondrial samples. As we stated in our previous response to reviewer #1 the ER is hard to detect in quiescent oocytes. So we switched our detection methods to obtain great sensitivity in our western blots and switch to our 3T3 cell model to test this possibility. As result we now show

that while we see signal for ER whole cell lysate (anti-KDEL) we do not observe any signal in our mitochondrial fractions. Moreover, using antibodies that target a plasma membrane protein DLL1 we find our mitochondrial fractions have no significant PM contamination. **Extended data figure 1B**

Additional response to reviewers.

In reference to point 2 the reviewer asks "I can see the full zymographs but not a blot for constitutive 20 staining for the full gel. I'm a bit worried that the alignment of the complexes is not correct as there is no active 20S present. Please provide a gel with activity and a full blot of this gel for a 20S subunit. This should then clearly show 26/30 and 20S complexes."

With regards to the reviewers concern about the 20S proteasome it is not surprising that the 20S proteasome shows no activity in native non-denaturing gels. Numerous papers in yeast and human cells show that if samples are prepared fresh and not frozen the 20S proteasome is latent. Only upon the addition of SDS or in situations where the proteasome is disrupted will the 20S proteasome show activity. Please see figure 2 in <https://pubmed.ncbi.nlm.nih.gov/21878651/>. I have attached the figure for your reference.

Typically, fresh well-prepared samples display very little 20S activity is present whereas frozen samples or samples of poor quality will have high levels of 20S activity due to breakdown of the 26S proteasome or artificial activation of latent 20S proteasome.

To further address the reviewers concern we have inserted, one of our groups zymography assays which also demonstrates this point by showing under normal condition no 20S activity is not detected. However, in response to INFgamma latent 20S is activated.

	Wt	KO	Wt	KO
IFN γ	-	-	+	+

LLVY-AMC

Regardless these zymography assays are supported by our AMC activity assays, ubiquitination assays, measurements of all three enzymatic activities of the proteasome, native westerns, and denaturing westerns.

We hope these points provide more clarity and resolve the concerns of the reviewer. We apologize not making these points clear in the previous draft.

REVIEWERS' COMMENTS

Reviewer #1 (Remarks to the Author):

The authors have adequately addressed my concerns.